# Particle-based Variational Inference with Preconditioned Functional Gradient Flow

**Hanze Dong[⋆], Xi Wang[‡], Yong Lin[†], Tong Zhang[⋆†]**
[⋆] Department of Mathematics, HKUST
[†] Department of Computer Science and Engineering, HKUST
[‡] College of Information and Computer Science, UMass Amherst

## Abstract

Particle-based variational inference (VI) minimizes the KL divergence between model samples and the target posterior with gradient flow estimates. With the popularity of Stein variational gradient descent (SVGD), the focus of particle-based VI algorithms has been on the properties of functions in Reproducing Kernel Hilbert Space (RKHS) to approximate the gradient flow. However, the requirement of RKHS restricts the function class and algorithmic flexibility. This paper offers a general solution to this problem by introducing a functional regularization term that encompasses the RKHS norm as a special case. This allows us to propose a new particle-based VI algorithm called *preconditioned functional gradient flow* (PFG). Compared to SVGD, PFG has several advantages. It has a larger function class, improved scalability in large particle-size scenarios, better adaptation to ill-conditioned distributions, and provable continuous-time convergence in KL divergence. Additionally, non-linear function classes such as neural networks can be incorporated to estimate the gradient flow. Our theory and experiments demonstrate the effectiveness of the proposed framework.

## 1 Introduction

Sampling from unnormalized density is a fundamental problem in machine learning and statistics, especially for posterior sampling. Markov Chain Monte Carlo (MCMC) (Welling & Teh, 2011; Hoffman et al., 2014; Chen et al., 2014) and Variational inference (VI) (Ranganath et al., 2014; Jordan et al., 1999; Blei et al., 2017) are two mainstream solutions: MCMC is asymptotically unbiased but sample-exhausted; VI is computationally efficient but usually biased. Recently, particle-based VI algorithms (Liu & Wang, 2016; Detommaso et al., 2018; Liu et al., 2019) tend to minimize the Kullback-Leibler (KL) divergence between particle samples and the posterior, and absorb the advantages of both MCMC and VI: (1) non-parametric flexibility and asymptotic unbiasedness; (2) sample efficiency with the interaction between particles; (3) deterministic updates. Thus, these algorithms are competitive in sampling tasks, such as Bayesian inference (Liu & Wang, 2016; Feng et al., 2017; Detommaso et al., 2018), probabilistic models (Wang & Liu, 2016; Pu et al., 2017).

Given a target distribution $p_*(x)$, particle-based VI aims to find $g(t, x)$, so that starting with $X_0 \sim p_0$, the distribution $p(t, x)$ of the following method: $dX_t = g(t, X_t)dt$, converges to $p_*(x)$ as $t \to \infty$. By the continuity equation (Jordan et al., 1998), we can capture the evolution of $p(t, x)$ by

$$\frac{\partial p(t, x)}{\partial t} = -\nabla \cdot (p(t, x)g(t, x)). \tag{1}$$

In order to measure the "closeness" between $p(t, \cdot)$ and $p_*$, we typically adopt the KL divergence,

$$D_{\mathrm{KL}}(t) = \int p(t, x) \ln \frac{p(t, x)}{p_*(x)} dx. \tag{2}$$

Using chain rule and integration by parts, we have

$$\frac{dD_{\mathrm{KL}}(t)}{dt} = -\int p(t, x)[\nabla \cdot g(t, x) + g(t, x)^\top \nabla_x \ln p_*(x)]dx, \tag{3}$$

which captures the evolution of KL divergence.

To minimize the KL divergence, one needs to define a "gradient" to update the particles as our $g(t, x)$. The most standard approach, *Wasserstein gradient* (Ambrosio et al., 2005), defines a gradient for $p(t, x)$ in the Wasserstein space, which contains probability measures with bounded second moments. In particular, for any functional $\mathcal{L}$ that maps probability density $p(t, x)$ to a non-negative scalar, we say that the particle density $p(t, x)$ follows the Wasserstein gradient flow of $\mathcal{L}$ if $g(t, x)$ is the gradient field of $L^2(\mathbb{R}^d)$-functional derivative of $\mathcal{L}$ (Villani, 2009). For KL divergence, the solution is $\nabla \ln \frac{p_*(x)}{p(t,x)}$. However, the computation of deterministic and time-inhomogeneous Wasserstein gradient is non-trivial. It is necessary to restrict the function class of $g(t, x)$ to obtain a tractable form.

Stein variational gradient descent (SVGD) is the most popular particle-based algorithm, which provides a tractable form to update particles with the kernelized gradient flow (Chewi et al., 2020; Liu, 2017). It updates particles by minimizing the KL divergence with a functional gradient measured in RKHS. By restricting the functional gradient with bounded RKHS norm, it has an explicit formulation: $g(t, x)$ can be obtained by minimizing Eq. (3). Nonetheless, there are still some limitations due to the restriction of RKHS: (1) the expressive power is limited because kernel method is known to suffer from the curse of dimensionality (Geenens, 2011); (2) with $n$ particles, the $O(n^2)$ computational overhead of kernel matrix is required. Further, we identify another crucial limitation of SVGD: the kernel design is highly non-trivial. Even in the simple Gaussian case, where particles start with $\mathcal{N}(0, I)$ and $p_* = \mathcal{N}(\mu_*, \Sigma_*)$, commonly used kernels such as linear and RBF kernel, have fundamental drawbacks in SVGD algorithm (Example 1).

Our motivation originates from functional gradient boosting (Friedman, 2001; Nitanda & Suzuki, 2018; Johnson & Zhang, 2019). For each $p(t, x)$, we find a proper function as $g(t, x)$ in the function class $\mathcal{F}$ to minimize Eq. (3). In this context, we design a regularizer for the functional gradient to approximate variants of "gradient" explicitly. We propose a regularization family to penalize the particle distribution's functional gradient output. For well-conditioned $-\nabla^2 \ln p_*$[1], we can approximate the Wasserstein gradient directly; For ill-conditioned $-\nabla^2 \ln p_*$, we can adapt our regularizer to approximate a preconditioned one. Thus, our functional gradient is an approximation to the preconditioned Wasserstein gradient. Regarding the function space, we do not restrict the function in RKHS. Instead, we can use non-linear function classes such as neural networks to obtain a better approximation capacity. The flexibility of the function space can lead to a better sampling algorithm, which is supported by our empirical results.

**Contributions.** We present a novel particle-based VI framework that incorporates functional gradient flow with general regularizers. We leverage a special family of regularizers to approximate the preconditioned Wasserstein gradient flow, which proves to be more effective than SVGD. The functional gradient in our framework explicitly approximates the preconditioned Wasserstein gradient, making it well-suited to handle ill-conditioned cases and delivering provable convergence rates. Additionally, our proposed algorithm eliminates the need for the computationally expensive $O(n^2)$ kernel matrix, resulting in increased computational efficiency for larger particle sizes. Both theoretical and empirical results demonstrate the superior performance of our framework and proposed algorithm.

## 2 ANALYSIS

**Notations.** In this paper, we use $x$ to denote particle samples in $\mathbb{R}^d$. The distributions are assumed to be absolutely continuous w.r.t. the Lebesgue measure. The probability density function of the posterior is denoted by $p_*$. $p(t, x)$ (or $p_t$) refers to particle distribution at time $t$. For scalar function $p(t, x)$, $\nabla_x p(t, x)$ denotes its gradient w.r.t. $x$. For vector function $g(t, x)$, $\nabla_x g(t, x)$, $\nabla_x \cdot g(t, x)$, $\nabla_x^2 g(t, x)$ denote its Jacobian matrix, divergence, Hessian w.r.t. $x$. $\frac{\partial p(t,x)}{\partial t}$ and $\frac{\partial g(t,x)}{\partial t}$ denotes the partial derivative w.r.t. $t$. Without ambiguity, $\nabla$ stands for $\nabla_x$ for conciseness. Notation $\|x\|_H^2$ stands for $x^\top H x$ and $\|x\|_I$ is denoted by $\|x\|$. Notation $\| \cdot \|_{\mathcal{H}^d}$ denotes the RKHS norm on $\mathbb{R}^d$.

We let $g(t, x)$ belong to a vector-valued function class $\mathcal{F}$, and find the best functional gradient direction. Inspired by the gradient boosting algorithm for regression and classification problems, we approximate the gradient flow by a function $g(t, x) \in \mathcal{F}$ with a regularization term, which solves the

---

[1] For any positive-definite matrix, the condition number is the ratio of the maximal eigenvalue to the minimal eigenvalue. A low condition number is well-conditioned, while a high condition number is ill-conditioned.

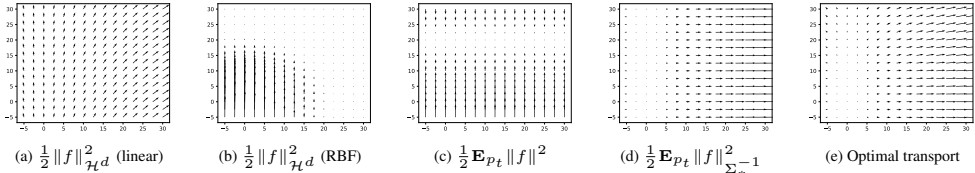

Figure 1: Illustration of the functional gradient $g(t, x)$ compared with optimal transport. (a)-(d) denotes the corresponding $Q$ of the regularized functional gradient. (a)-(b) are SVGD algorithms with linear and RBF kernel. Optimal transport denotes the direction of the shortest path towards $p_*$.

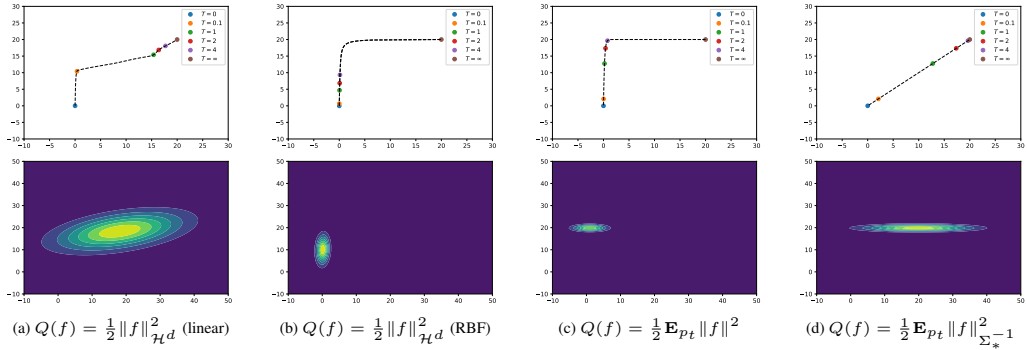

Figure 2: Evolution of particle distribution from $\mathcal{N}([0, 0]^\top, I)$ to $\mathcal{N}([20, 20]^\top, \text{diag}(100, 1))$ (first row: evolution of particle mean $\mu_t$; second row: particle distribution $p(5, x)$ at $t = 5$)

following minimization formulation:

$$g(t, x) = \arg\min_{f \in \mathcal{F}} \left[ -\int p(t, x)[\nabla \cdot f(x) + f(x)^\top \nabla \ln p_*(x)]dx + Q(f) \right], \quad (4)$$

where $Q(\cdot)$ is a regularization term that limit the output magnitude of $f$. This regularization term also implicitly determines the underlying "distance metric" used to define the gradient estimates $g(t, x)$ in our framework. When $Q(x) = \frac{1}{2} \int p(t, x)\|f(x)\|^2 dx$, Eq. (4) is equivalent to

$$g(t, x) = \arg\min_{f \in \mathcal{F}} \int p(t, x) \left\| f(x) - \nabla \ln \frac{p_*(x)}{p_t(x)} \right\|^2 dx. \quad (5)$$

If $\mathcal{F}$ is well-specified, i.e., $\nabla \ln \frac{p_*(x)}{p_t(x)} \in \mathcal{F}$, we have $g(t, x) = \nabla \ln \frac{p_*(x)}{p_t(x)}$, which is the direction of Wasserstein gradient flow. Interestingly, despite the computational intractability of Wasserstein gradient, Eq. (4) provides a tractable variational approximation.

For RKHS, we will show that SVGD is a special case of Eq. (4), where $Q(f) = \frac{1}{2}\|f\|^2_{\mathcal{H}^d}$ (Sec. 3.2.1). We point out that RKHS norm usually fails to regularize the functional gradient properly since it is fixed for any $p(t, x)$. Our next example shows that SVGD is a suboptimal solution for Gaussian case.

**Example 1.** *Consider that $p(t, \cdot)$ is $\mathcal{N}(\mu_t, \Sigma_t)$, $p_*$ is $\mathcal{N}(\mu_*, \Sigma_*)$. We consider the SVGD algorithm with linear kernel, RBF kernel, and regularized functional gradient formulation with $Q(f) = \frac{1}{2}\mathbf{E}_{p_t}\|f\|^2$, and $Q(f) = \frac{1}{2}\mathbf{E}_{p_t}\|f\|^2_{\Sigma_*^{-1}}$. Starting with $\mathcal{N}(0, I)$, Fig. 1 plots the $g(0, x)$ with different $Q$ and the optimal transport direction; the path of $\mu_t$ and $p(5, x)$ are illustrated in Fig. 2. The detailed mathematical derivations and analytical results are provided in Appendix A.2.*

Example 1 shows the comparison of different regularizations for the functional gradient. For RKHS norm, we consider the most commonly used kernels: linear and RBF. Fig. 1 shows the $g(0, x)$ of different regularizers: only $Q(f) = \frac{1}{2}\mathbf{E}_{p_t}\|f\|^2_{\Sigma_*^{-1}}$ approximates the optimal transport direction, while other $g(0, x)$ deviates significantly. SVGD with linear kernel underestimates the gradient of large-variance directions; SVGD with RBF kernel even suffers from gradient vanishing in low-density area. Fig. 2 demonstrates the path of $\mu_t$ with different regularizers. For linear kernel, due to the

curl component ($\mu_* \mu_t^\top$ term in linear SVGD is non-symmetric; See Appendix for details), $p(5, x)$ is rotated with an angle. For RBF kernel, the functional gradient of SVGD is not linear, leading to slow convergence. The $L_2$ regularizer is suboptimal due to the ill-conditioned $\Sigma_*$. We can see that $Q(f) = \frac{1}{2}\mathbf{E}_{p_t}\|f\|^2_{\Sigma_*^{-1}}$ produces the optimal path for $\mu_t$ (the line segment between $\mu_0$ and $\mu_*$).

## 2.1 GENERAL REGULARIZATION

Inspired by the Gaussian case, we consider the general form

$$Q(f(x)) = \frac{1}{2}\int p(t, x)\|f(x)\|^2_H dx \tag{6}$$

where $H$ is a symmetric positive definite matrix. Proposition 1 shows that with Eq. (6), the resulting functional gradient is the approximation of preconditioned Wasserstein gradient flow. It also implies the well-definedness and existence of $g(t, x)$ when $\mathcal{F}$ is closed, since $\|\cdot\|_H$ is lower bounded by 0.

**Proposition 1.** *Consider the case that $Q(f) = \frac{1}{2}\int p(t, x)\|f(x)\|^2_H dx$, where $H$ is a symmetric positive definite matrix. Then the functional gradient defined by Eq. (4) is equivalent as*

$$g(t, x) = \underset{f \in \mathcal{F}}{\arg\min} \frac{1}{2}\int p(t, x)\left\|f(x) - H^{-1}\nabla\ln\frac{p_*(x)}{p(t, x)}\right\|^2_H dx \tag{7}$$

**Remark.** Our regularizer is similar to the Bregman divergence in mirror descent. We can further extend $Q(\cdot)$ with a convex function $h(\cdot) : \mathbb{R} \to [0, \infty)$, where the regularizer is defined by $Q(f(x)) = \mathbf{E}_{p_t}h(f(x))$. We can adapt more complex geometry in our framework with proper $h(\cdot)$.

## 2.2 CONVERGENCE ANALYSIS

**Equilibrium condition.** To provide the theoretical guarantees, we show that the stationary distribution of our $g(t, x)$ update is $p_*$. Meanwhile, the evolution of KL-divergence is well-defined (without the explosion of functional gradient) and descending. We list the regularity conditions below.

[$\mathbf{A_1}$] (Regular function class) The discrepancy induced by function class $\mathcal{F}$ is positive-definite: for any $q \neq p$, there exists $f \in \mathcal{F}$ such that $\mathbf{E}_q[\nabla \cdot f(x) + f(x)^\top\nabla\ln p(x)] > 0$. For $f \in \mathcal{F}$ and $c \in \mathbb{R}$, $cf \in \mathcal{F}$, and $\mathcal{F}$ is closed. The tail of $f$ is regular: $\lim_{\|x\|\to\infty} f(\theta, x)p_*(x) = 0$ for $f \in \mathcal{F}$.

[$\mathbf{A_2}$] (*L*-Smoothness) For any $x \in \mathbb{R}^d$, $p_*(x) > 0$ and $p(t, x) > 0$ are *L*-smooth densities, *i.e.*, $\|\nabla\ln p(x) - \nabla\ln p(y)\| \leq L\|x - y\|$, with $\mathbf{E}_p\|x\|^2 < \infty$.

Particularly, [$\mathbf{A_1}$] is similar to the positive definite kernel in RKHS, which guarantees the decrease of KL divergence. [$\mathbf{A_2}$] is designed to make the gradient well-defined: RHS of Eq. (3) is finite.

**Proposition 2.** *Under* [$\mathbf{A_1}$], [$\mathbf{A_2}$], *when we update $X_t$ as Eq.* (7), *we have* $-\infty < \frac{dD_{\mathrm{KL}}}{dt} < 0$ *for all* $p(t, x) \neq p_*(x)$. *i.e.,* $g(t, x) = 0$ *if and only if* $p(t, x) = p_*(x)$.

Proposition 2 shows that the continuous dynamics of our $g(t, x)$ is well-defined and the KL divergence along the dynamics is descending. The only stationary distribution for $p(t, x)$ is $p_*(x)$.

**Convergence rate.** The convergence rate of our framework mainly depends on (1) the capacity of function class $\mathcal{F}$; (2) the complexity of $p_*$. In this section, we analyze that when the approximation error is small and the target $p_*$ is log-Sobolev, the KL divergence converges linearly.

[$\mathbf{A_3}$] ($\epsilon$-approximation) For any $t > 0$, there exists $f_t(x) \in \mathcal{F}$ and $\epsilon < 1$, such that

$$\int p(t, x)\left\|f_t(x) - H^{-1}\nabla\ln\frac{p(t, x)}{p_*(x)}\right\|^2_H dx \leq \epsilon\int p(t, x)\left\|\nabla\ln\frac{p(t, x)}{p_*(x)}\right\|^2_{H^{-1}} dx$$

[$\mathbf{A_4}$] The target $p_*$ satisfies $\mu$-log-Sobolev inequality ($\mu > 0$): For any differentiable function $g$, we have $\mathbb{E}_{p_*}\left[g^2\ln g^2\right] - \mathbb{E}_{p_*}\left[g^2\right]\ln\mathbb{E}_{p_*}\left[g^2\right] \leq \frac{2}{\mu}\mathbb{E}_{p_*}\left[\|\nabla g\|^2\right]$.

Specifically, [$\mathbf{A_3}$] is the error control of gradient approximation. With universal approximation theorem (Hornik et al., 1989), any continuous function can be estimated by neural networks, which indicates the existence of such a function class. [$\mathbf{A_4}$] is a common assumption in sampling literature. It is more general than strongly concave assumption (Vempala & Wibisono, 2019).

**Theorem 1.** *Under* $[\mathbf{A_1}]$-$[\mathbf{A_4}]$, *for any* $t > 0$, *we assume that the largest eigenvalue,* $\lambda_{\max}(H) = m$. *Then we have* $\frac{dD_{\mathrm{KL}}(t)}{dt} \le -\frac{1-\epsilon}{2} \mathbb{E}_{p_t} \left\| \nabla \ln \frac{p_t(x)}{p_*(x)} \right\|_{H^{-1}}^2$ *and* $D_{\mathrm{KL}}(t) \le \exp(-(1-\epsilon)\mu t/m) D_{\mathrm{KL}}(0)$.

Theorem 1 shows that when $p_*$ is log-Sobolev, our algorithm achieves linear convergence rate with a function class powerful enough to approximate the preconditioned Wasserstein gradient. Moreover, considering the discrete algorithm, we have to make sure that the step size is proper, *i.e.*, $\|H^{-1}\nabla^2 \ln p_*(x)\| \le c$, for some constant $c > 0$. Thus, the preconditioned $H = -c\nabla^2 \ln p_*$ is better than plain one $H = cLI$, since $(-\nabla^2 \ln p_*)^{-1} \succeq L^{-1}I$. For better understanding, we have provided a Gaussian case discretized proof in Appendix A.6 to illustrate this phenomenon.

## 3 PFG: PRECONDITIONED FUNCTIONAL GRADIENT FLOW

---
**Algorithm 1** PFG: Preconditioned Functional Gradient Flow
---
**Input:** Unnormalized target distribution $p_*(x) = e^{-U(x)}$, $f_\theta(x): \mathbb{R}^d \to \mathbb{R}^d$, initial particles (parameters) $\{x_0^i\}_{i=1}^n$, $\theta_0$, iteration parameter $T, T'$, step size $\eta, \eta'$, regularization function $h(\cdot)$.
**for** $t = 1, \ldots, T$ **do**
  Assign $\theta_t^0 = \theta_{t-1}$;
  **for** $t' = 1, \cdots, T'$ **do**
    Compute $\hat{L}(\theta) = \frac{1}{n} \sum_{i=1}^n \left( h(f_\theta(x_t^i)) + f_\theta(x_t^i) \cdot \nabla U(x_t^i) - \nabla \cdot f_\theta(x_t^i) \right)$
    Update $\theta_t^{t'} = \theta_t^{t'-1} - \eta' \nabla \hat{L}(\theta_t^{t'-1})$;
  **end**
  Assign $\theta_t = \theta_t^{T_1}$ and update particles $x_t^i = x_t^i + \eta \left( f_{\theta_t}(x_t^i) \right)$ for all $i = 1, \cdots, n$;
**end**
**Return:** Optimized particles $\{x_T^i\}_{i=1}^n$

---

### 3.1 ALGORITHM

We will realize our algorithm with parametric $f_\theta$ (such as neural networks) and discretize the update.

**Parametric Function Class.** We can let $\mathcal{F} = \{f_\theta(x) : \theta \in \Theta\}$ and apply $g(t, x) = f_{\hat{\theta}_t}(x)$, such that

$$\hat{\theta}_t = \underset{\theta \in \Theta}{\arg\min} \left[ \int p(t, x)[-\nabla \cdot f_\theta(x) - f_\theta(x)^\top \nabla \ln p_*(x) + \frac{1}{2}\|f_\theta(x)\|_H^2]dx \right], \quad (8)$$

where $H$ is a symmetric positive definite matrix estimated at time $t$. Eq. (8) is a direct result from Eq. (4) and (6). The parametric function class allows $f_\theta$ to be optimized by iterative algorithms.

**Choice of $H$.** Considering the posterior mean trajectory, it is equivalent to the conventional optimization, so that $-\nabla^2 \ln p_*$ is ideal (Newton's method) to implement discrete algorithms. We use diagonal Fisher information estimators for efficient computation as Adagrad (Duchi et al., 2011). We approximate the preconditioner $H$ for all particles at each time step $t$ by moving averaging.

**Discrete update.** We can present our algorithm by discretizing Eq. (4) and (8). Given $X_0 \sim p_0$, we update $X_k$ as $X_{k+1} = X_k + \eta f_{\hat{\theta}_k}(X_k)$, where $\hat{\theta}_k$ is obtained by Eq. (8) with (stochastic) gradient descent. The integral over $p(k, x)$ is estimated by particle samples. Full procedure is presented in Alg. 1, where the regularizer $h$ is $\frac{1}{2}\|\cdot\|_H^2$ by default.

### 3.2 COMPARISON WITH SVGD

#### 3.2.1 SVGD FROM A FUNCTIONAL GRADIENT VIEW

For simplicity, we prove the case under finite-dimensional feature map[2], $\psi(x): \mathbb{R}^d \to \mathbb{R}^h$. We assume that $\mathcal{F} = \{W\psi(x) : W \in \mathbb{R}^{d \times h}\}$, and let kernel $k(x, y) = \psi(x)^\top \psi(y)$, $Q(f) = \frac{1}{2}\|f\|_{\mathcal{H}^d}^2 = \frac{1}{2}\|W\|_F^2$, where RKHS norm is the Frobenius norm of $W$. The solution is defined by

$$\hat{W}_t = \underset{W \in \mathbb{R}^{d \times h}}{\arg\min} - \int p(t, x) \operatorname{trace}[W\nabla\psi(x) + W\psi(x)\nabla \ln p_*(x)^\top]dx + \frac{1}{2}\|W\|_F^2, \quad (9)$$

---
[2]Infinite-dimensional version is provided in Appendix A.7.

which gives $\hat{W}_t = \int p(t, x')[\nabla \psi(x')^\top + \nabla \ln p_*(x') \psi(x')^\top] dx'$. This implies that

$$g(t, x) = \hat{W}_t \psi(x) = \int p(t, y)[\nabla_y k(y, x) + \nabla \ln p_*(y) k(y, x)] dy, \tag{10}$$

which is equivalent to SVGD. For linear function classes, such as RKHS, $Q$ can be directly applied to the parameters, such as $W$ here. The regularization of SVGD is $\frac{1}{2}\|\cdot\|_F^2$ (the norm defined in RKHS). For non-linear function classes, such as neural networks, the RKHS norm cannot be defined.

### 3.2.2 LIMITATIONS OF SVGD

**Kernel function class.** As in Section 3.2.1, RKHS only contains linear combination of the feature map functions, which suffers from curse of dimensionality (Geenens, 2011). On the other hand, some non-linear function classes, such as neural network, performs well on high dimensional data (LeCun et al., 2015). The extension to non-linear function classes is needed for better performance.

**Gradient preconditioning in SVGD.** In Example 1, when $-\nabla^2 \ln p_*$ is ill-conditioned, PFG algorithm follows the shortest path from $\mu_0$ to $\mu_*$. Although SVGD can implement preconditioning matrices as (Wang et al., 2019), due to the curl component and time-dependent Jacobian of $d\mu_t/dt$, any symmetric matrix cannot provide optimal preconditioning (detailed derivation in Appendix).

**Suboptimal convergence rate.** For log-Sobolev $p_*$, SVGD with commonly used bounded smoothing kernels (such as RBF kernel) cannot reach the linear convergence rate (Duncan et al., 2019) and the explicit KL convergence rate is unknown yet. Meanwhile, the Wasserstein gradient flow converges linearly. When the function class is sufficiently large, PFG converges provably faster than SVGD.

**Computational cost.** For SVGD, main computation cost comes from the kernel matrix: with $n$ particles, we need $O(n^2)$ memory and computation. Our algorithm uses an iterative approximation to optimize $g(t, x)$, whose memory cost is independent of $n$ and computational cost is $O(n)$ (Bertsekas et al., 2011). The repulsive force between particles is achieved by $\nabla \cdot f$ operator on each particle.

## 4 EXPERIMENT

To validate the effectiveness of our algorithm, we have conducted experiments on both synthetic and real datasets. Without special declarations, we use parametric two-layer neural networks with Sigmoid activation as our function class. To approximate $H$ in real datasets, we use the approximated diagonal Hessian matrix $\hat{H}$, and choose $H = \hat{H}^\alpha$, where $\alpha \in \{0, 0.1, 0.2, 0.5, 1\}$; the inner loop $T'$ of PFG is chosen from $\{1, 2, 5, 10\}$, the hidden layer size is chosen from $\{32, 64, 128, 256, 512\}$. The parameters are chosen by validation. More detailed settings are provided in the Appendix.

**Gaussian Mixture.** To demonstrate the capacity of non-linear function class, we have conducted the Gaussian mixture experiments to show the advantage over linear function class (RBF kernel) with SVGD. We consider to sample from a 10-cluster Gaussian Mixture distribution. Both SVGD and our algorithm are trained with 1,000 particles. Fig. 3 shows that the estimated "score" by RBF kernel is usually unsatisfactory: (1) In low-density area, it suffers from gradient vanishing, which makes samples stuck at these parts (similar to Fig. 1 (b)); (2) The score function cannot distinguish connected clusters. Specifically, some clusters are isolated while others might be connected. The choice of bandwidth is hard. The fixed bandwidth makes the SVGD algorithm un-

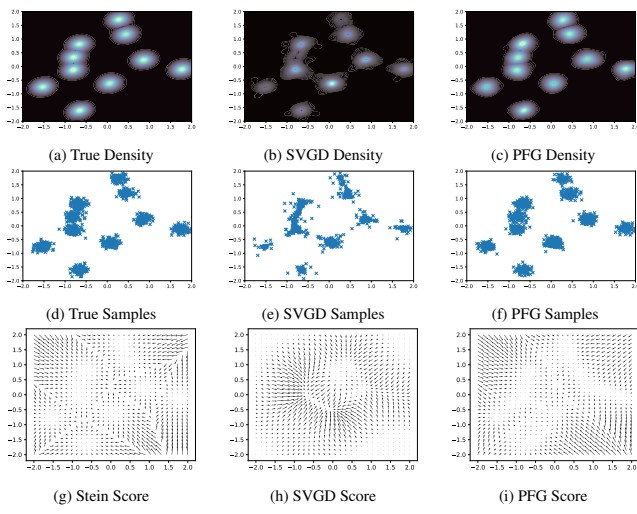

Figure 3: Particle-based VI for Gaussian mixture sampling.

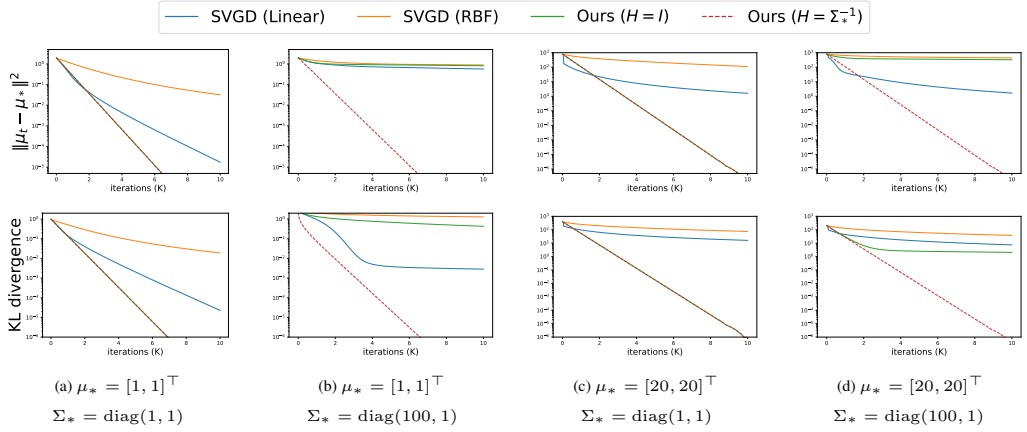

Figure 4: Evolution of particle distribution from $\mathcal{N}(0, I)$ to $\mathcal{N}(\mu_*, \Sigma_*)$ (first row: mean squared error of $\mu_t$: $\|\mu_t - \mu_*\|^2$; second row: KL divergence between $p(t, x)$ and $p_*(x)$)

able to determine the gradient near connected clusters. It fails to capture the clusters near $(-0.75, 0.5)$. However, the non-linear function class is able to resolve the above difficulties. In PFG, we found that the score function is well-estimated and the resulting samples mimic the true density properly.

**Ill-conditioned Gaussian distribution.** We show the effectiveness of our proposed regularizer. For ill-conditioned Gaussian distribution, the condition number of $\Sigma_*$ is large, *i.e.*, the ratio between maximal and minimal eigenvalue is large. We follow Example 1 and compare different $\mu_*$ and $\Sigma_*$. When $\Sigma_*$ is well-conditioned ($\Sigma_* = I$), $L_2$ regularizer (equivalent to Wasserstein gradient) performs well. However, it will be slowed down significantly with ill-conditioned $\Sigma_*$. For SVGD with linear kernel, the convergence slows down with shifted $\mu_*$ or ill-conditioned $\Sigma_*$. For SVGD with RBF kernel, the convergence is slow due to the improper function class. Interestingly, for ill-conditioned case, $\mu_t$ of SVGD (linear) converges faster than our method with $H = I$ but KL divergence does not always follow the trend. The reason is that $\Sigma_t$ of SVGD is highly biased, making KL divergence large. Our algorithm provides feasible Wasserstein gradient estimators and makes the particle-based sampling algorithm compatible with ill-conditioned sampling case.

**Logistic Regression.** We conduct Bayesian logistic regression for binary classification task on Sonar and Australian dataset (Dua & Graff, 2017). In particular, the prior of weight samples are assigned $\mathcal{N}(0, 1)$; the step-size is fixed as 0.01. We compared our algorithm with SVGD (200 particles), full batch gradient is used in this part. To measure the quality of posterior approximation, we use 3 metrics: (1) the distance between sample mean $\mu_t$ and posterior mean $\mu_*$; (2) the Maximum Mean Discrepancy (MMD) distance (Gretton et al., 2012) from particle samples to posterior samples; (3) the evidence lower bound (ELBO) of current particles, $\mathbf{E}_{x \sim p_t}[\log p_*(x) - \log p(t, x)]$, where $p_*(x)$ is the unnormalized density $p(x, \mathcal{D})$ with training data $\mathcal{D}$, the entropy term $\log p(t, x)$ is estimated by kernel density. The ground truth samples of $p_*$ is estimated by NUTS (Hoffman et al., 2014). Fig. 5 shows that our method outperforms SVGD consistently.

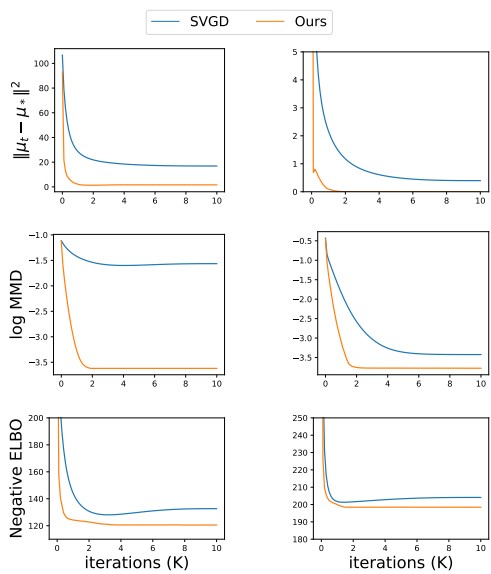

Figure 5: Posterior sampling for Bayesian logistic regression. (200 particles; dataset: *sonar* (first column) and *Australian* (second column). $\mu_t$: particle mean; $\mu_*$: posterior mean.)

**Hierarchical Logistic Regression.** For hierarchical logistic regression, we consider a 2-level hierarchical regression, where prior of weight samples is $\mathcal{N}(0, \alpha^{-1})$. The prior of $\alpha$ is

Gamma$(0, 0.01)$. All step-sizes are chosen from $\{10^{-t} : t = 1, 2, \cdots\}$ by validation. We compare the test accuracy and negative log-likelihood as Liu & Wang (2016). Mini-batch was used to estimate the log-likelihood of training samples (batch size is 200). Tab. 1 shows that our algorithm performs the best against SVGD and SGLD on hierarchical logistic regression.

**Bayesian Neural Networks.** We compare our algorithm with SGLD and SVGD variants on Bayesian neural networks. We use two-layer networks with 50 hidden units (100 for Year dataset, 128 for MNIST) and ReLU activation function; the batch-size is 100 (1000 for Year dataset), All step-sizes are chosen from $\{10^{-t} : t = 1, 2, \cdots\}$ by valida-tion. For UCI datasets, we choose the precondi-tioned version for comparison: M-SVGD (Wang et al., 2019) and pSGLD (Li et al., 2016). Data

Table 1: Hierarchical Logistic Regression on UCI datasets (particle size = 200)

|  | COVTYPE | | GERMAN | |
|---|---|---|---|---|
|  | ACC. | NLL | ACC. | NLL |
| SVGD | 75.1 | 0.58 | 65.1 | 0.71 |
| SGLD | 74.9 | 0.57 | 64.8 | 0.65 |
| PFG | **75.7** | **0.51** | **67.6** | **0.64** |

samples are randomly partitioned to two parts: 90% (training), 10% (testing) and 100 particles is used. For MNIST, data split follows the default setting and we compare our algorithm with SVGD for $5, 10, 50, 100$ particles (SVGD with 100 particles exceeded the memory limit). In Tab. 2, PFG outperforms SVGD and SGLD significantly. In Fig. 6, we found that the accuracy and NLL of PFG is better than SVGD with all particle size. With more particle samples, PFG algorithm also improves.

Table 2: Averaged test root-mean-square error (RMSE) and test log-likelihood of Bayesian Neural Networks on UCI datasets (100 particles). Results are computed by 10 trials.

| DATASET | AVERAGE TEST RMSE | | | AVERAGE TEST LOG-LIKELIHOOD | | |
|---|---|---|---|---|---|---|
|  | M-SVGD | pSGLD | PFG | M-SVGD | pSGLD | PFG |
| BOSTON | $2.72_{\pm 0.17}$ | $2.70_{\pm 0.16}$ | $\mathbf{2.47}_{\pm 0.11}$ | $-2.86_{\pm 0.20}$ | $-2.85_{\pm 0.18}$ | $\mathbf{-2.35}_{\pm 0.12}$ |
| CONCRETE | $4.83_{\pm 0.11}$ | $5.05_{\pm 0.13}$ | $\mathbf{4.69}_{\pm 0.14}$ | $-3.21_{\pm 0.06}$ | $-3.21_{\pm 0.07}$ | $\mathbf{-2.83}_{\pm 0.16}$ |
| ENERGY | $0.89_{\pm 0.10}$ | $0.99_{\pm 0.08}$ | $\mathbf{0.48}_{\pm 0.04}$ | $-1.42_{\pm 0.03}$ | $-1.31_{\pm 0.05}$ | $\mathbf{-1.22}_{\pm 0.06}$ |
| PROTEIN | $4.55_{\pm 0.14}$ | $4.59_{\pm 0.18}$ | $\mathbf{4.51}_{\pm 0.06}$ | $-3.07_{\pm 0.13}$ | $-3.22_{\pm 0.11}$ | $\mathbf{-2.89}_{\pm 0.07}$ |
| WINERED | $0.63_{\pm 0.04}$ | $0.64_{\pm 0.02}$ | $\mathbf{0.60}_{\pm 0.02}$ | $-1.77_{\pm 0.05}$ | $-1.80_{\pm 0.08}$ | $\mathbf{-1.61}_{\pm 0.03}$ |
| WINEWHITE | $0.65_{\pm 0.05}$ | $0.67_{\pm 0.07}$ | $\mathbf{0.59}_{\pm 0.02}$ | $-1.75_{\pm 0.04}$ | $-1.82_{\pm 0.07}$ | $\mathbf{-1.58}_{\pm 0.04}$ |
| YEAR | $8.62_{\pm 0.09}$ | $8.66_{\pm 0.07}$ | $\mathbf{8.56}_{\pm 0.04}$ | $-3.59_{\pm 0.08}$ | $-3.56_{\pm 0.04}$ | $\mathbf{-3.51}_{\pm 0.03}$ |

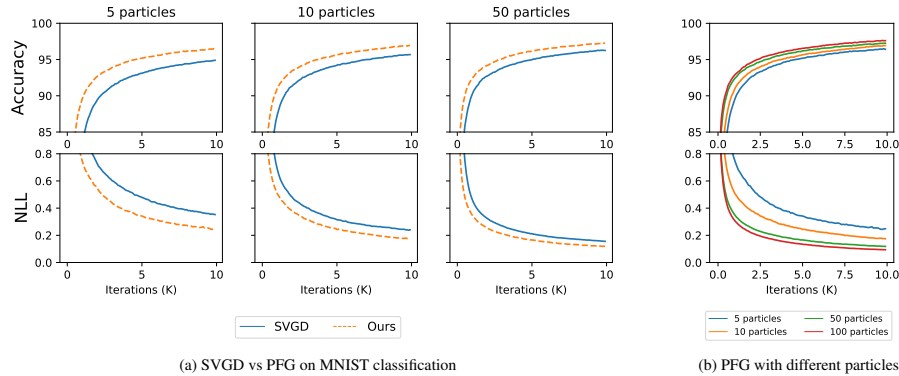

(a) SVGD vs PFG on MNIST classification

(b) PFG with different particles

Figure 6: Test accuracy and NLL of Bayesian Neural Network (MNIST classification)

**Time Comparison.** As indicated, our algorithm is more scalable than SVGD in terms of particle numbers. Tab. 3 shows that SVGD entails more computation cost with the increase of particle numbers. Our functional gradient is obtained with iterative approximation without kernel matrix computation. When $n$ is large, our $O(n)$ algorithm is much faster than $O(n^2)$ kernel-based method. Interestingly, the introduction of particle interactions is a key merit of particle-based VI algorithms, which intuitively needs $O(n^2)$ computation. The integration by parts technique draws the connection between the particle interaction and $\nabla \cdot f_\theta$ which supports more efficient computational realizations.

Table 3: Time Comparison on Bayesian logistic regression (sonar dataset, 1,000 iterations)

| # OF PARTICLES | 5 | 10 | 100 | 1000 | 2000 |
|---|---|---|---|---|---|
| SVGD | $5.20_{\pm 0.10}$ | $5.25_{\pm 0.13}$ | $5.71_{\pm 0.11}$ | $19.10_{\pm 0.52}$ | $81.50_{\pm 1.86}$ |
| PFG | $7.87_{\pm 0.16}$ | $7.95_{\pm 0.15}$ | $8.15_{\pm 0.17}$ | $11.56_{\pm 0.24}$ | $13.22_{\pm 0.26}$ |

## 5 RELATED WORKS

**Stein Discrepancy and SVGD.** Stein discrepancy (Liu et al., 2016; Chwialkowski et al., 2016; Liu & Wang, 2016) is known as the foundation of the SVGD algorithm, which is defined on the bounded function classes, such as functions with bounded RKHS norm. We provide another view to derive SVGD: SVGD is a functional gradient method with RKHS norm regularizer. From this view, we found that there are other variants of particle-based VI. We suggest that the our proposed framework has many advantages over SVGD as the function class is more general and the kernel computation can be saved. We use neural networks as our $\mathcal{F}$ in real practice, which is much more efficient than SVGD. Moreover, the general regularizer in Eq. (6) improves the conditioning of functional gradient, which is proven better than SVGD in both theory and experiments. The preconditioning of our algorithm is more explainable than SVGD, without incorporating RKHS-dependent properties.

**Learning Discrepancy with Neural Networks.** Some recent works (Hu et al., 2018; Grathwohl et al., 2020; di Langosco et al., 2021) leverage neural networks to learn the Stein discrepancy, which are related to our algorithm. They only implement $Q(f(x)) = \mathbf{E}_{p_t} \| f(x) \|^2$ to estimate the discrepancy. Grathwohl et al. (2020) measure the similarity between distributions and train energy-based models (EBM) with the discrepancy. However, they did not further validate the sampling performance of their "learned discrepancy", so it is a new version of KSD (Liu et al., 2016), rather than SVGD. Hu et al. (2018) train a neural "generator" for sampling tasks. They discussed the scaling of $L_2$ regularization to align with the step size. In contrast, we restrict our investigation in particle-based VI and avoid learning a "generator", which may introduce more parameterization errors. di Langosco et al. (2021) is an empirical study to implement the $L_2$ version of our algorithm with comparable performance to SVGD. We extend the design of $Q$ to a general case, and emphasize the benefits of our proposed regularizers corresponding to the preconditioned algorithm. We include the Example 1 and Fig. 4 to demonstrate the necessity of preconditioning. Our theoretical and experimental results have demonstrated the improvement of PFG algorithm against SVGD.

**KL Wasserstein Gradient Flow.** Wasserstein gradient flow (Ambrosio et al., 2005) is the continuous dynamics to minimize functionals in Wasserstein space, which is crucial in sampling and optimal transport tasks. However, the numerical computation of Wasserstein gradient is non-trivial. Previous works (Peyré, 2015; Benamou et al., 2016; Carlier et al., 2017) attempt to find tractable formulation with spatial discretization, which suffers from curse of dimensionality. More recently, (Mokrov et al., 2021; Alvarez-Melis et al., 2021) leverage neural networks to model the Wasserstein gradient and aim to find the full transport path between $p_0$ and $p_*$. The computation of full path is extremely large. Salim et al. (2020) defines proximal gradient in Wasserstein space by JKO operator. However, the work mainly focus on theoretical properties and the efficient implementation remains open. Wang et al. (2022) solves SDP to approximate Wasserstein gradient, which considers the dual form of the variational problem. When the functional is KL divergence, Wasserstein gradient can also be realized with Langevin dynamics (Welling & Teh, 2011; Bernton, 2018) by adding Gaussian noise to particles, which are also variants of MCMC. However, the deterministic algorithm to approximate Wasserstein gradient flow is still challenging. Our framework provides a tractable approximation to fix this gap.

## 6 CONCLUSION

In particle-based VI, we consider that the particles can be updated with a preconditioned functional gradient flow. Our theoretical results and Gaussian example led to an algorithm: PFG that can approximate preconditioned Wasserstein gradient directly. Our theory indicates that when the function class is large, PFG is provably faster than conventional particle-based VI, SVGD. The empirical result showed that PFG performs better than conventional SVGD and SGLD variants.

## ACKNOWLEDGEMENTS

The work was supported by the General Research Fund (GRF 16310222 and GRF 16201320).

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

## A  PROOFS

### A.1  PROOF OF EQ. (3)

From the continuity equation (or Fokker-Planck equation), we have

$$\frac{\partial p(t,x)}{\partial t} = -\nabla \cdot [g(t,x)p(t,x)]$$

By chain rule and integration by parts,

$$\begin{aligned}
\frac{dD_{\text{KL}(t)}}{dt} &= \int \frac{\partial p(t,x)}{\partial t} \left(1 + \ln \frac{p(t,x)}{p_*(x)}\right) dx \\
&= -\int \nabla \cdot [g(t,x)p(t,x)] \left(1 + \ln \frac{p(t,x)}{p_*(x)}\right) dx \\
&= \int [g(t,x)p(t,x)]^\top \nabla \ln \frac{p(t,x)}{p_*(x)} dx \\
&= \int [g(t,x)p(t,x)]^\top \nabla [\ln p(t,x) - \ln p_*(x)] dx \\
&= \int g(t,x)^\top [\nabla p(t,x) - p(t,x)\nabla \ln p_*(x)] dx \\
&= -\int p(t,x)[\nabla \cdot g(t,x) + g(t,x)\nabla_x \ln p_*(x)] dx.
\end{aligned}$$

### A.2  DISCUSSION OF EXAMPLE 1

(1) SVGD with linear kernel $k(x,y) = x^\top K y + 1$

$$g(t,x) = -\Sigma_*^{-1}[(\Sigma_t + (\mu_t - \mu_*)\mu_t^\top)Kx + \mu_t - \mu_*] + Kx. \tag{11}$$

(2) SVGD with RBF kernel $k_\sigma(x,y) = \frac{1}{\sqrt{\sigma^2}} \exp\left(-\frac{1}{2\sigma^2}\|y-x\|^2\right)$

$$g(t,x) = O(x\exp(-\|x\|^2)). \tag{12}$$

(3) Linear function class with $L_2$ regularization $Q(f(x)) = \frac{1}{2}\mathbf{E}_{p_t}\|f(x)\|^2$.

$$g(t,x) = -\Sigma_*^{-1}(x - \mu_*) + \Sigma_t^{-1}(x - \mu_t). \tag{13}$$

(4) Linear function class with Mahalanobis regularization $Q(f(x)) = \frac{1}{2}\mathbf{E}_{p_t}\|f(x)\|^2_{\Sigma_*^{-1}}$

$$g(t,x) = -(x - \mu_*) + \Sigma_* \Sigma_t^{-1}(x - \mu_t). \tag{14}$$

For optimal transport $g(t,x) = -(x - \mu_*) + \Sigma_t^{-1/2}(\Sigma_t^{1/2}\Sigma_*\Sigma_t^{1/2})^{1/2}\Sigma_t^{-1/2}(x - \mu_t)$.

Note that $\mu_*\mu_t^\top$ is not symmetric, which makes the distribution rotate. Figure 7 shows that we can split the curl component by Helmholtz decomposition, which means that this part would rotate the distribution and cannot be compromised by preconditioning method. Thus, we cannot find any $K$ to obtain a proper preconditioning.

*Proof.* We consider these 4 cases seperately.

(1) SVGD with linear kernel.

If $k(x,y) = x^\top K y + 1$, then $\nabla_y k(x,y) = Kx$,

$$\begin{aligned}
g(t,x) &= \int p(t,y)(-\Sigma_*^{-1}(y - \mu_*) + \Sigma_t^{-1}(y - \mu_t))(y^\top Kx + 1)dy \\
&= -\int p(t,y)(\Sigma_*^{-1}(y - \mu_*)(y^\top Kx + 1))dy \\
&\quad + \int p(t,y)(\Sigma_t^{-1}(yy^\top - \mu_t y^\top))dyKx \\
&= -\Sigma_*^{-1}[(\Sigma_t + (\mu_t - \mu_*)\mu_t^\top)Kx + \mu_t - \mu_*] + Kx
\end{aligned}$$

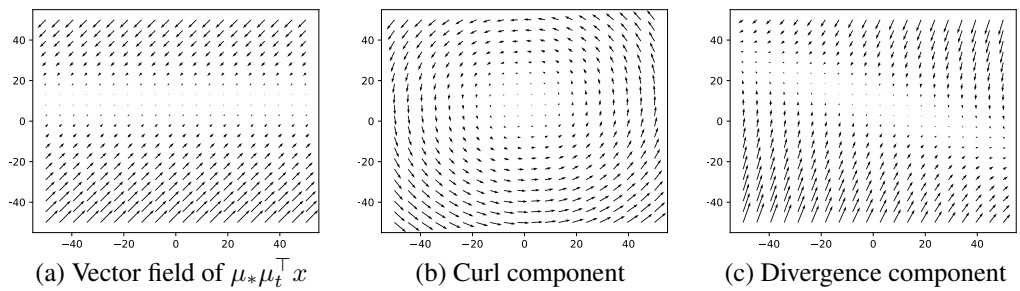

(a) Vector field of $\mu_*\mu_t^\top x$      (b) Curl component      (c) Divergence component

Figure 7: Illustration of $\mu_*\mu_t^\top$ term by Helmholtz decomposition. ($\mu_t = [0, 10], \mu_* = [20, 20]$)

(2) SVGD with RBF kernel.

$$
\begin{aligned}
g(t, x) &= \int \frac{1}{\sqrt{2\pi|\Sigma_t|}} \exp\left(-\frac{1}{2}(y - \mu_t)^\top \Sigma_t^{-1}(y - \mu_t)\right) \frac{1}{\sqrt{\sigma^2}} \exp\left(-\frac{1}{2\sigma^2}\|y - x\|^2\right) \\
&\quad \left(\Sigma_*^{-1}(y - \mu_*) - \Sigma_t^{-1}(y - \mu_t)\right) dy \\
&= \int \frac{1}{\sqrt{2\pi|\Sigma_t|}} \exp\left(-\frac{1}{2}\tilde{y}^\top \Sigma_t^{-1}\tilde{y}\right) \frac{1}{\sqrt{\sigma^2}} \exp\left(-\frac{1}{2\sigma^2}\|\tilde{y} + \mu_t - x\|^2\right) \\
&\quad \left(\Sigma_*^{-1}(\tilde{y} + \mu_t - \mu_*) - \Sigma_t^{-1}\tilde{y}\right) d\tilde{y} \\
&= \int \frac{\sqrt{2\pi|(\sigma^{-2}I + \Sigma_t^{-1})^{-1}|}}{\sqrt{2\pi\sigma^2|\Sigma_t|}} \frac{1}{\sqrt{2\pi|(\sigma^{-2}I + \Sigma_t^{-1})^{-1}|}} \cdot \\
&\quad \exp\left(-\frac{1}{2}(\tilde{y} - \mu_x)^\top \left(\sigma^{-2}I + \Sigma_t^{-1}\right)(\tilde{y} - \mu_x)\right) \cdot \\
&\quad \exp\left(\frac{1}{2}(\mu_t - x)^\top \Sigma_t^{-1}(\mu_t - x)\right)\left(\Sigma_*^{-1}(\tilde{y} + \mu_t - \mu_*) - \Sigma_t^{-1}\tilde{y}\right) d\tilde{y} \\
&= \int \frac{1}{\sqrt{\sigma^2|\Sigma_t(\sigma^{-2}I + \Sigma_t^{-1})|}} \tilde{p}(\tilde{y}) \exp\left(\frac{1}{2}(\mu_t - x)^\top \Sigma_t^{-1}(\mu_t - x)\right) \\
&\quad \left(\Sigma_*^{-1}(\tilde{y} + \mu_t - \mu_*) - \Sigma_t^{-1}\tilde{y}\right) d\tilde{y} \\
&= C_x \left(\left(\Sigma_*^{-1} - \Sigma_t^{-1}\right)\mu_x + \Sigma_*^{-1}(\mu_t - \mu_*)\right)
\end{aligned}
$$

where

$$
\mu_x = \sigma^{-2}\left(\sigma^{-2}I + \Sigma_t^{-1}\right)^{-1}(x - \mu_t) = \left(I + \sigma^2\Sigma_t^{-1}\right)^{-1}(x - \mu_t),
$$

$$
C_x = \frac{1}{\sqrt{|\sigma^2I + \Sigma_t|}} \exp\left(-\frac{1}{2\sigma^2}(x - \mu_t)^\top(I - (I + \sigma^2\Sigma_t^{-1})^{-1})(x - \mu_t)\right)
$$

Note that $g(t, x) = O(x \exp(-\|x\|^2))$. When $x \to \infty$, $g(t, x)$ is vanishing.

(3) and (4) are special case of Proposition 2.

Considering the evolution of $\mu_t$, we have

(1) $\frac{d\mu_t}{dt} = -\Sigma_*^{-1}[(\Sigma_t + (\mu_t - \mu_*)\mu_t^\top)K\mu_t + \mu_t - \mu_*] + K\mu_t$, which contains non-symmetric $\mu_*\mu_t^\top$.
(2) The transport is not linear.

(3) $\frac{d\mu_t}{dt} = \Sigma_*^{-1}(\mu_* - \mu_t)$,.

(4) $\frac{d\mu_t}{dt} = \mu_* - \mu_t$, which means that the evolution of (4) is equivalent with optimal transport.

$\square$

### A.3 PROOF OF PROPOSITION 1

*Proof.* Assume that $Q(f) = \int p(t,x) \|f(x)\|_H^2 dx$, when we have Eq. (4) as

$$g(t,x) = \underset{f \in \mathcal{F}}{\arg\min} \left[ \int p(t,x)[-\nabla \cdot f(x) - f(x)^\top \nabla \ln p_*(x) + \frac{1}{2}\|f(x)\|_H^2]dx \right] \tag{15}$$

$$= \underset{f \in \mathcal{F}}{\arg\min} \left[ \int p(t,x)[-f(x)^\top \nabla \ln \frac{p_*(x)}{p(t,x)} + \frac{1}{2}\|f(x)\|_H^2]dx \right] \tag{16}$$

$$= \underset{f \in \mathcal{F}}{\arg\min} \left[ \mathbf{E}_{p_t}[\frac{1}{2} \left\| H^{-1}\nabla \ln \frac{p_*(x)}{p(t,x)} \right\|_H^2 - f(x)^\top \nabla \ln \frac{p_*(x)}{p(t,x)} + \frac{1}{2}\|f(x)\|_H^2] \right] \tag{17}$$

$$= \underset{f \in \mathcal{F}}{\arg\min} \left[ \frac{1}{2}\mathbf{E}_{p_t} \left\| f(x) - H^{-1}\nabla \ln \frac{p_*(x)}{p(t,x)} \right\|_H^2 \right] \tag{18}$$

$\square$

### A.4 PROOF OF PROPOSITION 2

**Lemma 1.** *(A Variant of Lemma. 11 in (Vempala & Wibisono, 2019)) Suppose that $p(x) > 0$ is L-smooth. Then, we have*

$$\int p(x) \|\nabla \ln p(x)\|^2 \, dx \le Ld;$$

*Proof.* By $[\mathbf{A_1}]$, there exists $f$ such that

$$-\int p(t,x)[\nabla \cdot f + f^\top \nabla_x \ln p_*(x)]dx < 0. \tag{19}$$

Thus, consider $f_c(x) = cf(x)$,

$$L(c) = -\int p(t,x)[\nabla \cdot cf + cf^\top \nabla_x \ln p_*(x)]dx =: -C_f c \tag{20}$$

$$l(c) = \int p(t,x)\|cf\|_H^2 dx =: c^2 \int p(t,x)\|f\|_H^2 dx = C_H c^2 \tag{21}$$

where $C_H, C_f > 0$.

When $c = \frac{C_f}{2C_H}$, we have

$$C_H c^2 - C_f c = -\frac{C_f^2}{4C_H} < 0.$$

By choosing $c_0 = \frac{C_f}{2C_H}$,

$$-\int p(t,x)[\nabla \cdot g(t,x) + g(t,x)^\top \nabla \ln p_*(x) + \frac{1}{2}\|g(t,x)\|_H^2]dx \tag{22}$$

$$\le -\int p(t,x)[\nabla \cdot c_0 f + c_0 f^\top \nabla \ln p_*(x) + \frac{1}{2}\|c_0 f\|_H^2]dx \tag{23}$$

$$< 0 \tag{24}$$

Thus,

$$\frac{dD_{\mathrm{KL}}(t)}{dt} = -\int p(t,x)[\nabla \cdot g(t,x) + g(t,x)^\top \nabla \ln p_*(x)dx < 0$$

By Eq. (3),

$$\frac{dD_{\mathrm{KL}}(t)}{dt} = -\int p(t,x)[\nabla \cdot g(t,x) + g(t,x)^\top \nabla_x \ln p_*(x)]dx \tag{25}$$

$$= -\int p(t,x)[g(t,x)\nabla \ln \frac{p_*(x)}{p(t,x)}]dx \tag{26}$$

$$\geq -\frac{1}{2}\int p(t,x)\left\|\nabla \ln \frac{p_*(x)}{p(t,x)}\right\|^2 dx - \frac{1}{2}\int p(t,x)\left\|g(t,x)\right\|^2 dx \tag{27}$$

$$\geq -(1 + \lambda_{\max}(H)\lambda_{\min}^{-2}(H))\int p(t,x)\left\|\nabla \ln \frac{p_*(x)}{p(t,x)}\right\|^2 dx \tag{28}$$

$$\geq -(1 + \lambda_{\max}(H)\lambda_{\min}^{-2}(H))\left(Ld + \int p(t,x)\left\|\nabla \ln p_*(0) + Lx\right\|^2 dx\right) \tag{29}$$

$$> -\infty \tag{30}$$

where $\int p(t,x)\left\|\nabla \ln \frac{p_*(x)}{p(t,x)}\right\|^2 dx$ can be controlled by $Ld$ ($[\mathbf{A_2}]$ and Lemma 1) and $\mathbf{E}_{p_t}\|x\|^2$.

$\square$

### A.5 PROOF OF THEOREM 1

*Proof.* By $\mu$-log-Sobolev inequality in $[\mathbf{A_4}]$, suppose $g^2 = p_t/p_*$, then we have

$$D_{\mathrm{KL}}(t) \leq \frac{1}{2\mu}\int p(t,x)\left\|\nabla \ln \frac{p(t,x)}{p_*(x)}\right\|^2 dx. \tag{31}$$

When we consider the $H$-induced norm, for any $x$, we have $\|x\|_{H^{-1}} \geq \frac{1}{\lambda_{\max}(H)}\|x\|$. Thus,

$$D_{\mathrm{KL}}(t) \leq \frac{\lambda_{\max}(H)}{2\mu}\int p(t,x)\left\|\nabla \ln \frac{p(t,x)}{p_*(x)}\right\|_{H^{-1}}^2 dx. \tag{32}$$

Using Cauchy-Schwarz inequality and $[\mathbf{A_3}]$, we have

$$\frac{dD_{\mathrm{KL}}(t)}{dt} = -\int p(t,x)\nabla \ln \frac{p_*(x)}{p(t,x)} \cdot g(t,x)dx$$

$$= -\int p(t,x)\nabla \ln \frac{p_*(x)}{p(t,x)} \cdot \left(g(t,x) - H^{-1}\nabla \ln \frac{p_*(x)}{p(t,x)} + H^{-1}\nabla \ln \frac{p_*(x)}{p(t,x)}\right)dx$$

$$= -\int p(t,x)\left\|\nabla \ln \frac{p(t,x)}{p_*(x)}\right\|_{H^{-1}}^2 dx$$

$$+ \int p(t,x)\nabla \ln \frac{p_*(x)}{p(t,x)} \cdot \left(g(t,x) - H^{-1}\nabla \ln \frac{p_*(x)}{p(t,x)}\right)dx$$

$$\leq -\int p(t,x)\left\|\nabla \ln \frac{p(t,x)}{p_*(x)}\right\|_{H^{-1}}^2 dx$$

$$+ \int p(t,x)\left\|\nabla \ln \frac{p_*(x)}{p(t,x)}\right\|_{H^{-1}}\left\|g(t,x) - H^{-1}\nabla \ln \frac{p_*(x)}{p(t,x)}\right\|_H dx$$

$$\leq -\frac{1}{2}\int p(t,x)\left\|\nabla \ln \frac{p(t,x)}{p_*(x)}\right\|_{H^{-1}}^2 dx + \frac{1}{2}\int p(t,x)\left\|g(t,x) - H^{-1}\nabla \ln \frac{p_*(x)}{p(t,x)}\right\|_H^2 dx$$

$$\leq -\frac{\mu(1-\epsilon)}{\lambda_{\max}(H)}D_{\mathrm{KL}}(t)$$

Thus, by Grönwall's inequality,

$$D_{\mathrm{KL}}(t) \leq \exp\left(-\frac{\mu(1-\epsilon)}{\lambda_{\max}(H)}\right)D_{\mathrm{KL}}(0)$$

**Remark:** Theorem 1 shows that PFG algorithm convergence with linear rate. Moreover, the choice of preconditioning matrix affect the convergence rate with the largest eigenvalue. In real practice, due to the discretized algorithm, one needs to normalize the scaling of $H$, which is equivalent to the learning rate in practice. For any preconditioning $\tilde{H}$, we need to compute the corresponding learning rate to get $H = \eta^{-1}\tilde{H}$. One usually set that $\|\eta\tilde{H}^{-1}\nabla^2 \ln p_*(x)\|_2^{-1} \leq 1$ to make the discretized (first-order) algorithm feasible ($\eta \leq \|\tilde{H}^{-1}\nabla^2 \ln p_*(x)\|_2$), otherwise the second order error would be too large. The construction is similar to condition number dependency in optimization literature.

Take the fixed largest learning rate for example in the continuous case:

When $\tilde{H} = I$, we have that $\eta = 1/L$, $H = LI$,

$$\frac{dD_{\mathrm{KL}}(t)}{dt} \leq -\frac{1}{2L}\mathbb{E}_{p_t}\left\|\nabla \ln \frac{p_t(x)}{p_*(x)}\right\|^2.$$

When $\tilde{H} = -\nabla^2 \ln p_*$, we have that $\eta = 1$, $H = -\nabla^2 \ln p_*$,

$$\frac{dD_{\mathrm{KL}}(t)}{dt} \leq -\frac{1}{2}\mathbb{E}_{p_t}\left\|\nabla \ln \frac{p_t(x)}{p_*(x)}\right\|^2_{-(\nabla^2 \ln p_*)^{-1}} \leq -\frac{1}{2L}\mathbb{E}_{p_t}\left\|\nabla \ln \frac{p_t(x)}{p_*(x)}\right\|^2.$$

$\square$

### A.6 ANALYSIS OF DISCRETE GAUSSIAN CASE

As is shown in Example 1, given that $x_0 \sim \mathcal{N}(0, I)$ and the target distribution is $\mathcal{N}(\mu_*, \Sigma_*)$, we have the following update for our PFG algorithm,

$$x_{t+1} = x_t + \eta H^{-1}\left(\Sigma_*^{-1}(\mu_* - x_t) + \Sigma_t^{-1}(x_t - \mu_t)\right).$$

By taking the expectation, we have

$$\mu_{t+1} - \mu_t = \eta H^{-1}\left(\Sigma_*^{-1}(\mu_* - \mu_t)\right)$$
$$\Sigma_{t+1} - \Sigma_t = \eta H^{-1}\left(\Sigma_t^{-1} - \Sigma_*^{-1}\right)\Sigma_t + \eta\Sigma_t\left(\Sigma_t^{-1} - \Sigma_*^{-1}\right)H^{-1}$$
$$+ \eta^2 H^{-1}\left(\Sigma_t^{-1} - \Sigma_*^{-1}\right)\Sigma_t\left(\Sigma_t^{-1} - \Sigma_*^{-1}\right)H^{-1}$$

Since $\Sigma_0 = I$, the eigenvectors of $\Sigma_t$ will not change during the update. For notation simplicity, we write the diagonal case in the following discussion.

Assume that $\Sigma_* = \mathrm{diag}(\sigma_1^2, \cdots, \sigma_d^2)$, $\Sigma_t = \mathrm{diag}(s_1^2(t), \cdots, s_d^2(t))$, $H = \mathrm{diag}(h_1, \cdots, h_d)$, such that $\sigma_i \geq \sigma_{i+1}$ for any $i = 1, \cdots, d-1$. Without loss of generality, we assume that $\sigma_1 \leq 1$, otherwise we can still rescale the initialization to obtain the similar constraint.

Given the fact that $x - \ln(1 + x) \leq \frac{x^2}{2}$ when $x > 0$, we have,

$$D_{\mathrm{KL}}(t) \leq \frac{1}{2}\left(\|\mu_t - \mu_*\|^2_{\Sigma_*^{-1}} + \frac{1}{2}\sum_{i=1}^{d}\left(\frac{s_i^2(t) - \sigma_i^2}{\sigma_i^2}\right)^2\right).$$

Thus, we define

$$C_0 := \frac{1}{2}\left(\|\mu_0 - \mu_*\|^2_{\Sigma_*^{-1}} + \frac{1}{2}\sum_{i=1}^{d}\left(\frac{1 - \sigma_i^2}{\sigma_i^2}\right)^2\right). \tag{33}$$

Considering the $i$-th dimension (with notation $[\cdot]_i$),

$$[\mu_{t+1}]_i = \left(1 - \frac{\eta}{h_i\sigma_i^2}\right)[\mu_t]_i + \frac{\eta}{h_i\sigma_i^2}[\mu_*]_i \tag{34}$$

$$s_i^2(t+1) = s_i^2(t) + \frac{2\eta s_i^2(t)}{h_i}\left(\frac{1}{s_i(t)^2} - \frac{1}{\sigma_i^2}\right) + \frac{\eta^2 s_i^2(t)}{h_i^2}\left(\frac{1}{s_i(t)^2} - \frac{1}{\sigma_i^2}\right)^2. \tag{35}$$

To guarantee the convergence of the algorithm, we need to make sure that both $\mu_t$ and $\Sigma_t$ converge. By rearranging (34) and (35), we have

$$\frac{[\mu_{t+1}]_i - [\mu_*]_i}{\sigma_i} = \left(1 - \frac{\eta}{h_i \sigma_i^2}\right) \frac{[\mu_t]_i - [\mu_*]_i}{\sigma_i}$$

$$\frac{s_i^2(t+1) - \sigma_i^2}{\sigma_i^2} = \left(1 - \frac{2\eta}{h_i \sigma_i^2}\right) \left(\frac{s_i^2(t) - \sigma_i^2}{\sigma_i^2}\right) + \frac{\eta^2}{h_i^2 \sigma_i^2 s_i^2(t)} \left(\frac{s_i^2(t) - \sigma_i^2}{\sigma_i^2}\right)^2.$$

One needs to construct contraction sequences of both mean and variance gap, which are guided by $1 - \eta/(h_i \sigma_i^2)$ and $1 - 2\eta/(h_i \sigma_i^2)$. Notice that $s_i^2(0) \geq \sigma_i^2$. Assume that $\eta \leq \min_i h_i \sigma_i^2/2$. Then we have $s_i^2(t+1) \geq \sigma_i^2$.

Thus, we obtain

$$\frac{\eta}{h_i} \left| \frac{1}{\sigma_i^2} - \frac{1}{s_i^2(t)} \right| \leq 1.$$

It is natural to obtain the contraction of both mean and variance gap,

$$\|\mu_{t+1} - \mu_*\|_{\Sigma_*^{-1}}^2 \leq \max_i \left(1 - \frac{\eta}{h_i \sigma_i^2}\right)^2 \|\mu_t - \mu_*\|_{\Sigma_*^{-1}}^2$$

$$\left| \frac{s_i^2(t+1) - \sigma_i^2}{\sigma_i^2} \right| \leq \left(1 - \frac{\eta}{h_i \sigma_i^2}\right) \left| \frac{s_i^2(t) - \sigma_i^2}{\sigma_i^2} \right|$$

$$D_{\mathrm{KL}}(t) \leq \max_i \left(1 - \frac{\eta}{h_i \sigma_i^2}\right)^{2t} C_0,$$

where $C_0$ is defined in Eq. (33).

When $h_i = 1$, we have $\eta = \frac{\sigma_d^2}{2}$

$$D_{\mathrm{KL}}(t) \leq \left(1 - \frac{\sigma_d^2}{2\sigma_1^2}\right)^{2t} C_0.$$

When $h_i = \sigma_i^{-2}$, we have $\eta = \frac{1}{2}$

$$D_{\mathrm{KL}}(t) \leq \frac{1}{4^t} C_0.$$

This result is equivalent to the Remark in A.5.

### A.7 PROOF OF INFINITE-DIMENSIONAL CASE OF RKHS

*Proof.* Assume the feature map, $\psi(x) : \mathbb{R}^d \to \mathcal{H}$ and let kernel $k(x, y) = \langle \psi(x), \psi(y) \rangle_{\mathcal{H}}$, $Q(f) = \frac{1}{2}\|f\|_{\mathcal{H}^d}$. One can perform spectral decomposition to obtain that

$$k(x, y) = \sum_{i=1}^{\infty} \lambda_i \psi_i(x) \psi_i(y)$$

where $\psi_i : \mathbb{R}^d \to \mathbb{R}$ are orthonormal basis and $\lambda_i$ is the corresponding eigenvalue.

For any $g \in \mathcal{H}^d$, we have the decomposition,

$$g(x) = \sum_{i=1}^{\infty} g_i \sqrt{\lambda_i} \psi_i(x),$$

where $g_i \in \mathbb{R}^d$ and $\sum_{i=1}^{\infty} \|g_i\|^2 < \infty$.

The solution is defined by

$$\hat{g} = \underset{g \in \mathcal{H}^d}{\arg\min} - \int p(t, x) \left( \nabla \cdot g + \nabla \ln p_*(x)^\top g(x) \right) + \frac{1}{2} \|g\|_{\mathcal{H}^d}^2, \tag{36}$$

$$= \underset{g \in \mathcal{H}^d}{\arg\min} - \int p(t, x) \left( \nabla \cdot \sum_{i=1}^{\infty} g_i \sqrt{\lambda_i} \psi_i(x) + \sum_{i=1}^{\infty} \sqrt{\lambda_i} \nabla \ln p_*(x)^\top g_i \psi_i(x) \right) + \sum_{i=1}^{\infty} \|g_i\|^2, \tag{37}$$

which gives

$$\hat{g}_i = \sqrt{\lambda_i} \int p(t, y) [\nabla \psi_i(y) + \nabla \ln p_*(y) \psi_i(y)] dy.$$

This implies that

$$g(t, x) = \sum_{i=1}^{\infty} \sqrt{\lambda_i} \hat{g}_i \psi_i(x) = \int p(t, y) [\nabla_y k(y, x) + \nabla \ln p_*(y) k(y, x)] dy, \tag{38}$$

which is equivalent to SVGD.

$\square$

## B  MORE DISCUSSIONS

### B.1  SCORE MATCHING

Score matching (SM) is related to our algorithm due to the integration by parts technique, which uses parameterized models to estimate the stein score $\nabla \ln p_*$. We made some modifications to the original techniques in SM: (1) We extend the score matching $\nabla \ln p_*$ to Wasserstein gradient approximation $\nabla \ln p_* - \nabla \ln p_t$; (2) We introduce the geometry-aware regularization to approximate the preconditioned gradient. Thus, our proposed modifications make it suitable for sampling tasks.

In a word, both SM and PFG are derived from the integration by parts technique. SM is a very promising framework in generative modeling and our PFG is more suitable for sampling tasks.

### B.2  OTHER PRECONDITIONED SAMPLING ALGORITHMS

Preconditioning is a popular topic in SVGD literature. Stein variational Newton method (SVN) (Detommaso et al., 2018) approximates the Hessian to accelerate SVGD. However, due to the gap between SVGD and Wasserstein gradient flow, the theoretical interpretation is not clear. Matrix SVGD (Wang et al., 2019) leverages more general matrix-valued kernels in SVGD, which includes SVN as a variant and is better than SVN. (Chen et al., 2019) perform a parallel update of the parameter samples projected into a low-dimensional subspace by an SVN method. It implements dimension reduction to SVN algorithm. Liu et al. (2022) projects SVGD onto arbitrary dimensional subspaces with Grassmann Stein kernel discrepancy, which implement Riemannian optimization technique to find the proper projection space.

Note that all of these algorithms are based on RKHS norm regularization. As we mentioned in Example 1, due to the introduction of RKHS, the preconditioning matrix is often hard to find. For example, the preconditioning matrix of the linear kernel varies in time, while our algorithm only needs a constant matrix. For RBF kernel, the previous works have demonstrated the improvement of the Hessian inverse matrix, but the design is still heuristic due to the gap between the Wasserstein gradient and SVGD. Our algorithm directly approximates the preconditioned Wasserstein gradient, and further analysis is clear and conclusive. Also, the SVGD-based algorithms suffer from the drawbacks of RKHS norm regularization, as discussed in Example 1.

Besides, there are some other interesting works (Li et al., 2019; Garbuno-Inigo et al., 2020; Wang et al., 2021; Lin et al., 2021; Wang & Li, 2020; 2022; Li & Ying, 2019) also take the preconditioning methods / local geometry / subspace properties into consideration, and they have also shown the improvement over the plain versions, which have similar motivation to our algorithm. Although these methods are out of the particle-based VI scope, we believe that these methods have great potentials in our literature, which can be interesting future works.

### B.3  OTHER WASSERSTEIN GRADIENT FLOW ALGORITHMS

There is another line of work to approximate Wasserstein gradient flow through JKO-discretization (Mokrov et al., 2021; Alvarez-Melis et al., 2021; Fan et al., 2021). The continuous versions of these Wasserstein gradient flow algorithms are related to our algorithm when the functional is chosen as KL divergence. These are promising alternative algorithms to PFG in transport tasks. From the theoretical view, they solves the JKO operator with special neural networks (ICNN) and aims to estimate the Wasserstein gradient of general functionals, including KL/JS divergence, etc. On the other hand, our algorithm is motivated by the continuous case of KL Wasserstein gradient flow and we only consider the Euler discretization (rather than JKO), which is the setting of particle-based variational inference.

When we consider the KL divergence and posterior sampling task, our proposed method is more efficient than Wasserstein gradient flow based ones, due to the flexibility of the gradient estimation function. We provide the empirical results on Bayesian logistic regression below (Covtype dataset).

Table 4: Bayesian logistic regression (Covtype dataset)

| METHOD | ACCURACY (%) | NLL |
|---|---|---|
| OURS | **75.7** | **0.58** |
| JKO-ICNN | 75.0 | 0.62 |
| JKO-VARIATIONAL | 75.3 | 0.59 |

According to the table, our algorithm outperforms JKO-type Wasserstein gradient methods. Moreover, it is important to mention that JKO-type Wasserstein gradient is extremely computation-exhausted, due to the computation of JKO operator. Thus, it would be more challenging to apply them to larger tasks such as Bayesian neural networks.

### B.4 VARIANCE COLLAPSE AND HIGH-DIMENSIONAL PERFORMANCE

Some recent works (Ba et al., 2021; Gong et al., 2020; Zhuo et al., 2018; Liu et al., 2022) suggest that SVGD suffers from curse of dimensionality and the variance of SVGD-trained particles tend to diminish in high-dimensional spaces. We highlight that one of the key issues is that the SVGD algorithm with RBF function space is improper. As shown in Tab. 5, when fitting a Gaussian distribution, an ideal function class is the linear function class given the Gaussian initialization. However, provided with RBF-based SVGD, the variance collapse phenomenon is extremely severe. On the contrary, when using a proper function class (linear), both SVGD and PFG performs well. More importantly, for some powerful non-linear function classes (neural networks), it still performs well with PFG. The effectiveness of neural network function class and PFG algorithm would be particularly important when the target distribution is more complex.

Table 5: Illustration of Variance Collapse in RBF function classes (20-dim $\mathcal{N}(0, I)$)

| METHOD | TARGET | SVGD (RBF) | SVGD (LINEAR) | PFG (LINEAR) | PFG (NEURAL NETWORK) |
|---|---|---|---|---|---|
| VARIANCE | 1.00 | 0.35 | 1.01 | 1.00 | 0.99 |

To further justify the effectiveness of PFG algorithm in high dimensional cases, we evaluate our model under different settings following Liu et al. (2022). Tab. 6 shows that our PFG algorithm is robust to high dimensional fitting problems, which is comparable to GSVGD (Liu et al., 2022) in Gaussian case.

Table 6: Estimating the dimension-averaged marginal variance of $\mathcal{N}(0, I)$

| DIMENSION | 20 | 40 | 60 | 80 | 100 |
|---|---|---|---|---|---|
| SVGD | 0.35 | 0.18 | 0.12 | 0.09 | 0.04 |
| GSVGD | 0.96 | 0.97 | **1.00** | 1.01 | **1.00** |
| PFG | **1.00** | **0.99** | 0.98 | **1.00** | 0.97 |

We also include a further justification in Tab. 7, which compute the energy distance between particle the target distribution and the particle estimation on a 4-mode Gaussian mixture distribution. In this case, the function class of gradient should be much larger than linear case. We can find that the resulting performance improves the previous work (GSVGD).

Table 7: Energy distance ($\times 10^{-2}$) between the target distribution and the particle estimation on Multimodal Gaussian mixture distribution (4 modes)

| DIMENSION | 20 | 40 | 60 | 80 | 100 |
|---|---|---|---|---|---|
| SVGD | 2.1 | 6.7 | 10.8 | 24.9 | 39.6 |
| GSVGD | 1.0 | 2.3 | 3.4 | 3.2 | 3.5 |
| PFG | **0.6** | **0.8** | **1.2** | **1.6** | **2.1** |

## B.5 ESTIMATION OF PRECONDITIONING MATRIX

Ideally, one would leverage the exact Hessian inverse as the preconditioning matrix to make the algorithm well-conditioned, which is also the target of other preconditioned SVGD algorithms. However, the computation of exact Hessian is challenging. Thus, motivated by adaptive gradient optimization algorithms, we leverage the moving average of Fisher information to obtain the preconditioning matrix. We conduct hierarchical logistic regression to discuss the importance of preconditioning. We compare 3 kinds of preconditioning strategies to estimate: (1) Full Hessian: compute the exact full Hessian matrix; (2) K-FAC: Kronecker-factored Approximate Curvature to estimate Hessian; (3) Diagonal: Diagonal moving average to estimate Hessian. According to the table, we can find that the preconditioning indeed improves the performance of Bayesian inference, and the choice of estimation strategies does not differ much. Thus, we choose the most efficient one: diagonal estimation.

Table 8: Average Performance on Hierarchical Logistic Regression (German dataset)

| | FULL HESSIAN | K-FAC | DIAGONAL | W/O PRECONDITIONING |
|---|---|---|---|---|
| ACCURACY | 67.8 | 67.4 | 67.6 | 66.2 |
| NLL | 0.62 | 0.65 | 0.64 | 0.70 |

## B.6 MORE COMPARISONS

We also include a more complete empirical results on Bayesian neural networks, including the comparison with plain version of SVGD (Liu & Wang, 2016), SGLD (Welling & Teh, 2011), and Sliced SVGD (S-SVGD) (Gong et al., 2020). Our algorithm is still the most competitive one according to the table. Besides, we also conduct an ablation study to further justify the importance of preconditioning, which implies the value of $Q(\cdot)$ design. In Table 11, we have shown that the full algorithm with proper $Q$ outperforms the plain version.

Table 9: Comparison with plain version of SVGD and SGLD. Averaged test root-mean-square error (RMSE) and test log-likelihood of Bayesian Neural Networks on UCI datasets (100 particles). Results are computed by 10 trials.

| DATASET | AVERAGE TEST RMSE | | | AVERAGE TEST LOG-LIKELIHOOD | | |
|---|---|---|---|---|---|---|
| | SVGD | SGLD | PFG | SVGD | SGLD | PFG |
| BOSTON | $3.04_{\pm 0.12}$ | $2.79_{\pm 0.16}$ | $\mathbf{2.47}_{\pm 0.11}$ | $-2.76_{\pm 0.15}$ | $-2.63_{\pm 0.14}$ | $\mathbf{-2.35}_{\pm 0.12}$ |
| CONCRETE | $5.51_{\pm 0.17}$ | $4.97_{\pm 0.12}$ | $\mathbf{4.69}_{\pm 0.14}$ | $-3.07_{\pm 0.23}$ | $-3.03_{\pm 0.13}$ | $\mathbf{-2.83}_{\pm 0.16}$ |
| ENERGY | $1.96_{\pm 0.10}$ | $0.84_{\pm 0.07}$ | $\mathbf{0.48}_{\pm 0.04}$ | $-2.15_{\pm 0.11}$ | $-1.82_{\pm 0.16}$ | $\mathbf{-1.22}_{\pm 0.06}$ |
| PROTEIN | $4.89_{\pm 0.11}$ | $4.61_{\pm 0.13}$ | $\mathbf{4.51}_{\pm 0.06}$ | $-3.12_{\pm 0.14}$ | $-2.99_{\pm 0.14}$ | $\mathbf{-2.89}_{\pm 0.07}$ |
| WINERED | $0.68_{\pm 0.03}$ | $0.67_{\pm 0.05}$ | $\mathbf{0.60}_{\pm 0.02}$ | $-1.92_{\pm 0.05}$ | $-1.81_{\pm 0.08}$ | $\mathbf{-1.61}_{\pm 0.03}$ |
| WINEWHITE | $0.69_{\pm 0.04}$ | $0.71_{\pm 0.06}$ | $\mathbf{0.59}_{\pm 0.02}$ | $-1.85_{\pm 0.04}$ | $-1.69_{\pm 0.07}$ | $\mathbf{-1.58}_{\pm 0.04}$ |
| YEAR | $8.65_{\pm 0.08}$ | $8.69_{\pm 0.06}$ | $\mathbf{8.56}_{\pm 0.04}$ | $-3.54_{\pm 0.10}$ | $-3.64_{\pm 0.09}$ | $\mathbf{-3.51}_{\pm 0.03}$ |

Table 10: Comparison with Sliced SVGD (S-SVGD). Averaged test root-mean-square error (RMSE) and test log-likelihood of Bayesian Neural Networks on UCI datasets (100 particles). Results are computed by 10 trials.

| DATASET | AVERAGE TEST RMSE | | AVERAGE TEST LL | |
|---|---|---|---|---|
| | S-SVGD | PFG | S-SVGD | PFG |
| BOSTON | $2.87_{\pm0.16}$ | $\mathbf{2.47}_{\pm0.11}$ | $-2.50_{\pm0.16}$ | $\mathbf{-2.35}_{\pm0.12}$ |
| CONCRETE | $4.88_{\pm0.08}$ | $\mathbf{4.69}_{\pm0.14}$ | $-3.00_{\pm0.02}$ | $\mathbf{-2.83}_{\pm0.16}$ |
| ENERGY | $1.13_{\pm0.05}$ | $\mathbf{0.48}_{\pm0.04}$ | $-1.54_{\pm0.05}$ | $\mathbf{-1.22}_{\pm0.06}$ |
| PROTEIN | $4.58_{\pm0.07}$ | $\mathbf{4.51}_{\pm0.06}$ | $-2.99_{\pm0.12}$ | $\mathbf{-2.89}_{\pm0.07}$ |
| WINERED | $0.64_{\pm0.01}$ | $\mathbf{0.60}_{\pm0.02}$ | $-1.82_{\pm0.06}$ | $\mathbf{-1.61}_{\pm0.03}$ |
| WINEWHITE | $0.62_{\pm0.01}$ | $\mathbf{0.59}_{\pm0.02}$ | $-1.71_{\pm0.03}$ | $\mathbf{-1.58}_{\pm0.04}$ |
| YEAR | $8.77_{\pm0.06}$ | $\mathbf{8.56}_{\pm0.04}$ | $-3.60_{\pm0.05}$ | $\mathbf{-3.51}_{\pm0.03}$ |

Table 11: Ablation Study. Averaged test root-mean-square error (RMSE) and test log-likelihood of Bayesian Neural Networks on UCI datasets (100 particles). Results are computed by 10 trials.

| DATASET | AVERAGE TEST RMSE | | AVERAGE TEST LL | |
|---|---|---|---|---|
| | W/O PRECONDITION | PFG (FULL) | W/O PRECONDITION | PFG (FULL) |
| BOSTON | $2.69_{\pm0.13}$ | $\mathbf{2.47}_{\pm0.11}$ | $-2.55_{\pm0.12}$ | $\mathbf{-2.35}_{\pm0.12}$ |
| CONCRETE | $4.90_{\pm0.06}$ | $\mathbf{4.69}_{\pm0.14}$ | $-2.94_{\pm0.02}$ | $\mathbf{-2.83}_{\pm0.16}$ |
| ENERGY | $1.15_{\pm0.05}$ | $\mathbf{0.48}_{\pm0.04}$ | $-1.58_{\pm0.08}$ | $\mathbf{-1.22}_{\pm0.06}$ |
| PROTEIN | $4.60_{\pm0.08}$ | $\mathbf{4.51}_{\pm0.06}$ | $-3.04_{\pm0.04}$ | $\mathbf{-2.89}_{\pm0.07}$ |
| WINERED | $0.66_{\pm0.02}$ | $\mathbf{0.60}_{\pm0.02}$ | $-1.75_{\pm0.06}$ | $\mathbf{-1.61}_{\pm0.03}$ |
| WINEWHITE | $0.64_{\pm0.02}$ | $\mathbf{0.59}_{\pm0.02}$ | $-1.80_{\pm0.03}$ | $\mathbf{-1.58}_{\pm0.04}$ |
| YEAR | $8.67_{\pm0.06}$ | $\mathbf{8.56}_{\pm0.04}$ | $-3.59_{\pm0.06}$ | $\mathbf{-3.51}_{\pm0.03}$ |

## B.7 PARTICLE-BASED VI VS LANGEVIN DYNAMICS

One may be interested in the reason why we choose particle-based VI rather than Langevin dynamics. In the continuous case, Langevin dynamics solves Fokker-Planck equation. We have several reasons to demonstrate the superiority of proposed framework rather than Langevin dynamics.

*1. Motivation of particle-based variational inference: Deterministic update and repulsive interactions.*

One of the key algorithmic differences between particle-based variational inference and Langevin dynamics is the realization of $\nabla \ln p_t$, where particle-based variational inference explicitly estimates the deterministic repulsive function and Langevin dynamics uses Brownian motion.

The deterministic version repulsive force introduces interactions between particles while the stochastic version only maintain the variance with the randomness. Thus, for each particle, only the deterministic algorithm leads to a convergence, which is more stable and efficient (wrt the number of particles).

Figure 8 has demonstrated the phenomenon: we use both particle-based variational inference and Langevin dynamics to sample from Gaussian distribution. It is clear that although both algorithms return reasonable particles, the Langevin-induced particles are fully random and highly unstable due to the empirical randomness, but particle-based VI is robust against different random seeds. When the number of particles is small, the sample efficiency of particle-based VI should be much better than Langevin dynamics.

Besides, the deterministic update can induce a transport function, which can be used to map input to the target distribution directly, which can be a great potential of our propose framework. For example, we can maintain the composite function of all the particle updates. When $x_{t+1} = f_t(x_t) = x_t + \eta g(t, x_t)$, $t = 1, \cdots, T-1$, then $x_T = f_{T-1} \circ \cdots \circ f_1(x_1)$, then we can perform resampling painlessly. For the Gaussian case, this composite function is just a linear transform without other overheads. For neural networks, we may also use distillation to obtain a transport function. We believe that the distillation of the transport function have great potential in the future.

*2. Discretization of Fokker-Planck equation: Forward-Flow discretization vs Euler discretization.*

About the discretization of Fokker-Planck equation, Langevin dynamics is performing Forward-Flow (FFl) discretization, which is not the same as conventional gradient desent (Euler discretization). In a word, FFI only discretizes the $\ln p_*$ term, but solve $\ln p_t$ term with SDE, so that the discretized gradient is biased in general (Wibisono, 2018). The particle-based variational inference tend to perform Euler discretization, which is unbiased by discretizing the full (Wasserstein) gradient (similar to conventional gradient descent). Thus, from a theoretical perspective, Euler-type discretization is simpler and more direct, which is worthwhile to be further explored as our paper.

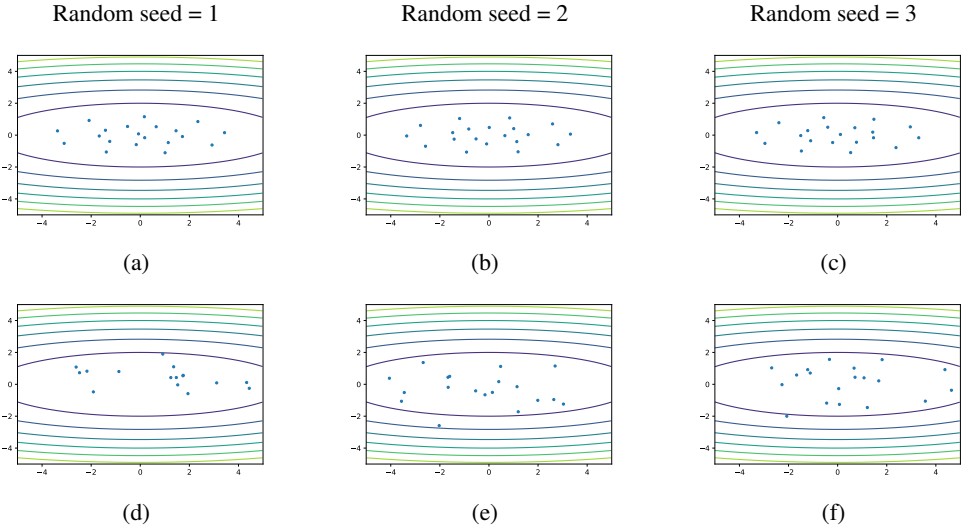

Figure 8: Particle-based VI (a-c) vs Langevin dynamics (d-f). (Blue dots refer to particles obtained)

*3. Function classes: non-linear function class (neural networks) vs linear function class (RKHS). (Note that the linearity is wrt function bases rather than the plain linear function)*

In our framework, both RKHS and neural networks (or other function class) are valid function classes to estimate the Wasserstein gradient.

In particular, neural networks are proven to outperform conventional RKHS in many areas, because it is a non-linear function class with learnable features, that can work with uneven subspaces. The RBF kernel uses the same smoothing operator for all gradients. In Figure 3, we have shown that the RKHS is incapable of capturing functional gradient near connected clusters, while neural networks can do so. As a result, the sampling quality can be improved.

## C    IMPLEMENTATION DETAILS

All experiments are conducted on Python 3.7 with NVIDIA 2080 Ti. Particularly, we use PyTorch 1.9 to build models. Besides, Numpy, Scipy, Sklearn, Matplotlib, Pillow are used in the models.

### C.1    SYNTHETIC EXPERIMENTS

#### C.1.1    ILL-CONDITIONED GAUSSIAN DISTRIBUTION

For ill-conditioned Gaussian distribution, we reproduce the continuous dynamics with Euler discretization. We consider 4 cases: (a) $\mu_* = [1, 1]^\top$, $\Sigma_* = \text{diag}(1, 1)$; (b) $\mu_* = [1, 1]^\top$, $\Sigma_* = \text{diag}(100, 1)$; (c) $\mu_* = [20, 20]^\top$, $\Sigma_* = \text{diag}(1, 1)$; (d) $\mu_* = [20, 20]^\top$, $\Sigma_* = \text{diag}(100, 1)$ to illustrate the behavior of SVGD and our algorithm clearly.

Notably, for our algorithms and SVGD (linear kernel), we use step size $10^{-3}$ to approximate the continuous dynamics and solve the mean and variance exactly (equivalent to infinite particles); for SVGD with RBF kernel, we use step size $10^{-2}$ with $1,000$ particles to approximate mean and variance. For the neural network function class, we assume that the function class of two-layer neural network $\mathcal{F}_\theta = \{f_\theta : \theta \in \Theta\}$. In practice, we may incorporate base shift $f_0$ as gradient boosting to accelerate the convergence, we use $\mathcal{F} = \{f_0 + f_\theta : f_\theta \in \mathcal{F}_\theta\}$, where $f_0(x) = 0$ or $f_0(x) = c\nabla \ln p_*(x)$ with some constant $c$. In practice, we select $c$ from $\{0, 0.1, 0.2, 0.5, 1\}$ by validation. Empirically, high dimensional models (such as Bayesian neural networks) can be accelerated significantly with the base function.

#### C.1.2    GAUSSIAN MIXTURE

For Gaussian Mixture, we sample a 2-$d$ 10-cluster Gaussian Mixture, where cluster means are sampled from standard Normal distribution, covariance matrix is $0.1^2 I$, the marginal probability of each cluster is $1/10$. The function class is 2-layer network with 32 hidden neurons and Tanh activation function. For inner loop, we choose $T' = 5$ and SGD optimizer (lr = 1e-3) with momentum 0.9. For particle optimization, we choose step-size = 1e-1. For SVGD, we choose RBF kernel with median bandwidth, step-size is 1e-2.

### C.2    APPROXIMATE POSTERIOR INFERENCE

To compute the trace, we use the randomized trace estimation (Hutchinson's trace estimator) (Hutchinson, 1989) to accelerate the computation .

- Sample $\{\xi_i\}_{i=1}^K$, such that $\mathbf{E}\xi = 0$ and $\mathbf{Cov}\xi = I$;
- Compute $\hat{L}(\theta) = \frac{1}{m}\sum_{i=1}^m \left(Q(f_\theta(x_t^i) + f_0(x_t^i)) - f_\theta(x_t^i) \cdot \nabla U(x_t^i) - \frac{1}{K}\sum_{j=1}^K \xi_i^\top \nabla f_\theta(x_t^i)\xi_i\right)$;
- Compute $\hat{L}(\theta) = \frac{1}{m}\sum_{i=1}^m \left(Q(f_\theta(x_t^i) + f_0(x_t^i)) - f_\theta(x_t^i) \cdot \nabla U(x_t^i) - \nabla \cdot f_\theta(x_t^i)\right)$ ;
- Update $\theta_t^{t'} = \theta_t^{t'-1} - \eta' \nabla \hat{L}(\theta_t^{t'-1})$;

where $\xi$ is chosen as Rademacher distribution.

#### C.2.1    LOGISTIC REGRESSION

For logistic regression, we compare our algorithm with SVGD in terms of the "goodness" of particle distribution. The ground truth is computed by NUTS (Hoffman et al., 2014). The mean is computed by 40,000 samples, the MMD is computed with 4,000 samples. The $H$ in this part is chosen as $H = I$ or $H = \hat{H}^{0.1}$.

In the experiments, we select step-size from $\{10^{-1}, 10^{-2}, 10^{-3}, 10^{-4}\}$ for each algorithm by validation. And we use Adam optimizer without momentum. For SVGD, we choose RBF kernel with median bandwidth. For PFG, we select inner loop from $\{1, 2, 5, 10\}$ by validation. The hidden neuron is 32.

### C.2.2 HIERARCHICAL LOGISTIC REGRESSION

For hierarchical logistic regression, we use the test performance to measure the different algorithms, including likelihood and accuracy. The $H$ in this part is chosen as $H = \hat{H}^{0.5}$.

For all experiments, batch-size is 200 and we select step-size from $\{10^{-1}, 10^{-2}, 10^{-3}, 10^{-4}\}$ for each algorithm by validation. And we use Adam optimizer without momentum. For SVGD, we choose RBF kernel with median bandwidth. For PFG, we select inner loop from $\{1, 2, 5, 10\}$ by validation. The hidden neuron is 32.

### C.2.3 BAYESIAN NEURAL NETWORKS (BNN)

We have two experiments in this part: UCI datasets, MNIST classification. The metric includes MSE/accuracy and likelihood. The hidden layer size is chosen from $\{32, 64, 128, 256, 512\}$. To approximate $H$, we use the approximated diagonal Hessian matrix $\hat{H}$, and choose $H = \hat{H}^\alpha$, where $\alpha \in \{0, 0.1, 0.2, 0.5, 1\}$. The inner loop $T'$ of PFG is chosen from $\{1, 2, 5, 10\}$ and with SGD optimizer (lr = 1e-3, momentum=0.9).

For UCI datasets, we follow the setting of SVGD (Liu & Wang, 2016). Data samples are randomly partitioned to two parts: 90% (training), 10% (testing) and we use 100 particles for inference. We conduct the two-layer BNN with 50 hidden units (100 for Year dataset) with ReLU activation function. The batch-size is 100 (1000 for Year dataset). Step-size is 0.01.

For MNIST, we conduct the two-layer BNN with 128 neurons. We compared the experiments with different particle size. Step-sizes are chosen from $\{10^{-1}, 10^{-2}, 10^{-3}, 10^{-4}\}$. The batch-size is 100.

