# OpenReview forum: "Particle-based Variational Inference with Preconditioned Functional Gradient Flow"
_ICLR.cc/2023/Conference — ICLR 2023 poster_

### Official Review · Reviewer_vc1L · 2022-10-16

**Confidence:** 4
**Correctness:** 3
**Technical Novelty And Significance:** 2
**Empirical Novelty And Significance:** 3
**Recommendation:** 6

**Clarity, Quality, Novelty And Reproducibility:**

The paper is clear and well written. However, learning functional gradient flows for particle-based VI is not new, and similar ideas (Hu et al., 2018; Grathwohl et al., 2020) have been exploited eariler. More specifically, this paper is highly related to a previous workshop paper: neural variational gradient descent (Lauro Langosco et al., 2021). Despite this, the theoretical analysis part and precondition trick is new.

**Strength And Weaknesses:**

Strength:

1. The precondition functional gradient flow for particle-based VI seems to be new.
2. The method has been well motivated with clear theoretical justification.
3. The advantages over SVGD has been clearly stated and illustrated in the experiments.


Weaknesses:

1. The preconditioning seems to be crucial for performance, lack of ablation study except for the toy examples.
2. No theoretical analysis for the precondition matrix H.
3. Discretization error of ODE is not explicitly considered.



**Summary Of The Paper:**

This paper proposed a new partical-based variational inference method by minimization the functional gradient of KL with a regularization where preconditioning seems to be crucial for the performance. The efficiency compared to SVGD is illustrated on several standard benchmark data sets. Relations to previous work were also discussed.

**Summary Of The Review:**

The paper proposed a new precondition functional gradient flow approach for particle-based VI and demonstrated its efficiency on several benchmark problems against SVGD. The method can be viewed as a score matching approach that learns the discrepancy of the target score and current particle scores, and update the particles according to the learned functional gradient flows. In that sense, the paper is highly related to a recent paper called neural variational gradient descent by Lauro Langosco et al., 2021. Despite this, the paper do provide some theoretical analysis regarding convergence given well trained functional gradients and introduce the precondition trick which seems to be crucial for performance. I, therefore, recommend it for a weak acceptance.


minor issue: there is a mistake in the upper bound of dKL/dt term in Theorem 1.

---

> ### Author Response · Authors · 2022-11-09
> **Response to Reviewer vc1L**
>
> Thanks for your suggestions and your insightful suggestion can indeed improve our work.
>
>
> **Q1: Ablation study.**
>
> Thanks for your suggestion. We provide the BNN experiment as below.
> We have shown the importance of preconditioning and we will incorperate them in our Appendix.
>
>
> *Average Test RMSE comparisons on Bayesian Neural Network*
> | Data  | w/o precondition | PFG (full) |
> | :---: |  :---: |    :---: |
> | Boston |  2.69$_{\pm0.13}$ |   **2.47**$_{\pm0.11}$|
> | Concrete |  4.90$_{\pm0.06}$ |   **4.69**$_{\pm0.14}$|
> | Energy |  1.15$_{\pm0.05}$ |   **0.48**$_{\pm0.05}$|
> | Protein |  4.60$_{\pm0.08}$ |   **4.51**$_{\pm0.05}$|
> | WineRed | 0.66$_{\pm0.02}$ |   **0.60**$_{\pm0.02}$|
> | WineWhite |  0.64$_{\pm0.02}$ |   **0.59**$_{\pm0.02}$|
> | Year |  8.67$_{\pm0.06}$ |   **8.56**$_{\pm0.04}$|
>
>
> *Average Test Log-Likelihood comparisons on Bayesian Neural Network*
> | Data  | w/o precondition | PFG (full) |
> | :---: |  :---: |    :---: |
> | Boston |  -2.55$_{\pm0.12}$ |  **-2.35**$_{\pm0.12}$|
> | Concrete |  -2.94$_{\pm0.02}$ |   **-2.83**$_{\pm0.16}$|
> | Energy |  -1.58$_{\pm0.08}$ |    **-1.22**$_{\pm0.06}$|
> | Protein |  -3.04$_{\pm0.04}$ |   **-2.89**$_{\pm0.05}$|
> | WineRed | -1.75$_{\pm0.06}$ |    **-1.61**$_{\pm0.03}$|
> | WineWhite |  -1.80$_{\pm0.03}$ |    **-1.58**$_{\pm0.04}$|
> | Year |  -3.59$_{\pm0.06}$ |   **-3.51**$_{\pm0.03}$|
>
>
>
> **Q2: Precondition matrix H.**
>
> Thank you and we will highlight our efforts in our paper.
> We have demonstrated that the preconditioning is particularly important in Gaussian case, which leads to faster convergence in both theory and experiments (Example 1, Appendix A.2,  A.6). For general case, we have incorperate the $H$ into our theory and demonstated the effect of $H$ choice in Theorem 1. We believe that our theoretical insight and practicle experiments have demonstrated the necessity of preconditioning.
>
>
>
> **Q3: Discretization error of ODE.**
>
> We provide the continuous-time analysis to demonstrate the property of our algorithm in ideal cases. The comparison between the approximated Wasserstein gradient flow and SVGD clearly demonstrates the advantage.
> We have to highlight that the rigorous proof of Wasserstein gradient flow with Euler discretization is **highly non-trivial** since the regularity of $p_t$ is usually not controllable, especially for the misspecified estimation. To the best of our knowledge, the Euler-type discretized analysis (of Wasserstein gradient flow) is an open problem yet. It would be inspiring to work on it as future work. And we have proposed a tractable way to perform the Euler-type discretization with general geometry (defined by $Q$) and do believe that our work has already shed light on this field.
>
> **Q4: Neural variational gradient descent (NVGD).**
>
> Thank you for your reference. We will cite this paper in our latest version.
> As you mentioned, our paper includes more theoretical results and preconditioning and we'd like to point out that the gains are based on a systematic framework, not just tricks.
>
> First, our framework is a general, which includes (1) SVGD; (2) $Q(x) = \frac{1}{2}\Vert x\Vert_2^2$; (3) $Q(x) = \frac{1}{2}\Vert x\Vert_H^2$, etc. But NVGD is only a special case of our algorithm. Our framework clearly demonstrated the differences ($Q(x)$ design) among our algorithm, NVGD and SVGD, while NVGD is from a heuristic perspective. Moreover, our framework does not restrict the function class to be neural networks, and other proper choices are also possible (such as linear function class, and even some RKHS).
>
> Second, our analysis of Gaussian case (linear function class) has demonstrated the importance of $Q(x)$ design, which clarifies the limitations of SVGD and NVGD and induces our algorithm smoothly.
>
> Third, the general design (the preconditioner to refine the geometry) makes our algorithm more suitable for high dimensional cases such as BNN experiments, which also coincides with the empirical experience in conventional optimization.
>
> We call attention to the community that the regularization $Q(\cdot)$ induced functional gradient is computationally tractable and the design of $Q$ is important for particle-based VI (determines SVGD or NVGD). Besides, we propose an excellent choice as our PFG algorithm. We believe that the discussion of other $Q(\cdot)$ can be interesting future works, such as other Bregman divergences.
>
> **Q5:Typos.**
>
> Thank you very much. We have fixed them in the latest version.

---

### Official Review · Reviewer_m3yE · 2022-10-25

**Confidence:** 4
**Correctness:** 4
**Technical Novelty And Significance:** 2
**Empirical Novelty And Significance:** 2
**Recommendation:** 8

**Clarity, Quality, Novelty And Reproducibility:**

This paper is well-organized and easy to read. The proof technique is rather straightforward.

Question:

- Could you elaborate on why $-c \nabla^2 \log p_*$ is better preconditioning? A bit detailed explanation of this would be helpful.
- Is there a connection between Assumption A3 and weak learning conditions for boosting methods?

**Strength And Weaknesses:**

**Strengths**:

The proposed method can be seen as a proper extension of Stein variational gradient descent (SVGD). Indeed, SVGD is a special case of the proposed method through the explicit feature representation of the kernel. In this sense, this work makes a certain contribution to the context by proposing a general form of SVGD.

**Weaknesses**:

The motivation for using neural networks for representing the drift term is somewhat unclear. Indeed, Langevin dynamics can be directly applied for sampling from Gibbs distribution $p_*$ and this dynamics does not require the approximation. (Note that Langevin dynamics also solves Fokker-Planck equation Eq. (1) with the Wasserstein gradient.) Thus, I think the standard Langevin dynamics is sufficiently powerful in terms of optimization. Moreover, there are also preconditioned variants of Langevin dynamics:

1. Max Welling and Yee Whye Teh. Bayesian Learning via Stochastic Gradient Langevin Dynamics.
2. Gaétan Marceau-Caron and Yann Ollivier. Natural Langevin Dynamics for Neural Networks.
3. C. Li et al., Preconditioned Stochastic Gradient Langevin Dynamics for Deep Neural Networks.
4. A. Wibisono. Proximal Langevin Algorithm: Rapid Convergence Under Isoperimetry

These preconditioned Langevin dynamics can be competitors to the proposed method. Thus, the discussion of these methods would be helpful. In particular,  [WIbisono] (see Section 4.1) considered the dynamics which optimizes Eq. (7) in the submission.

**Summary Of The Paper:**

This paper proposes a new variational inference method for sampling from the distribution. The proposed method is a sort of extension of the SVGD method inspired by functional gradient boosting. That is, the drift term is modeled by the neural networks and is optimized to maximize the discrepancy under an appropriate regularization. This work also provides a convergence guarantee under the assumptions regarding the approximation error to the preconditioned Wasserstein gradient. The effectiveness of the method is empirically verified on several tasks: sampling from Gaussian mixture, ill-conditioned Gaussian distribution, and Bayesian and hierarchical logistic regressions.

**Summary Of The Review:**

The proposed method is interesting because it is a proper extension of SVGD. But, the motivation for using neural networks as the drift term is rather weak. More discussion to motivate the use of the method will make the paper stronger.

---

> ### Author Response · Authors · 2022-11-09
> **Response to Reviewer m3yE (Part 1/2)**
>
> Thanks for your suggestions and your insightful suggestion can indeed improve our work.
>
> **Q1: The motivation for using neural networks for representing the drift term.**
>
> Thank you for the good question. We have several reasons to demonstrate the superiority of our proposed framework.
>
> *1. Motivation of particle-based variational inference: Deterministic update and repulsive interations*
>
> One of the key algorithmic differences between particle-based variational inference and Langevin dynamics is the realization of $\nabla \ln p_t$, where particle-based variational inference explicitly estimates the deterministic repulsive function and Langevin dynamics uses Brownian motion.
>
> The deterministic version (repulsive force) introduces interactions between particles, while the stochastic version maintains the variance with additional randomness. Thus, for each particle, only the deterministic algorithm leads to an exact convergence, which is more stable and sample-efficient (wrt the number of particles) in sampling algorithms.
>
> You can refer to Appendix B7 for an illustration. We have shown that the particle-based variational inference is more robust when the number of particles is small.
>
> Besides, the deterministic update can induce a transport function, which can be used to map input particles to the target distribution directly. This would be a great potential of our proposed framework. For example, we can maintain the composite function of all the particle updates. When $x_{t+1} = f_t(x_t)= x_t+\eta g(t,x_t)$, $t=1,\cdots, T-1$, then $x_T = f_{T-1}\circ\cdots\circ f_1(x_1)$, then we can perform resampling painlessly. For the Gaussian case, this composite function is just a linear transform without other overheads. For neural networks, we may also use distillation to obtain a transport function. We believe that the investigation of the transport function distillation has great potential in the future.
>
>
> *2. Discretization of Fokker-Planck equation: Forward-Flow discretization vs Euler discretization.*
>
> About the discretization of Fokker-Planck equation, Langevin dynamics is performing Forward-Flow (FFl) discretization, which is not the same as conventional gradient descent (Euler discretization). In a word, FFI only discretizes the $\ln p_*$ term, but solve $\ln p_t$ term with SDE, so that the discretized gradient is biased in general [Wibisono 2018]. Metropolis-Hastings correction is an ad-hoc way to resolve the issue, but the rejection rate can also be high in some cases. The particle-based variational inference tend to perform Euler discretization, which is unbiased by discretizing the full (Wasserstein) gradient (similar to conventional gradient descent).   Thus, from a theoretical perspective, Euler-type discretization is unbiased, natural, and more direct, which is worthwhile to be further explored as our paper.
>
> [Wibisono 2018] Wibisono, Andre. "Sampling as optimization in the space of measures: The Langevin dynamics as a composite optimization problem." Conference on Learning Theory. PMLR, 2018.
>
> *3. Function classes: non-linear function class (neural networks) vs linear function class (RKHS). (Note that the linearity is wrt function bases rather than the plain linear function)*
>
> In our framework, both RKHS and neural networks (or other function classes) are valid function classes to estimate the Wasserstein gradient.
>
> In particular, neural networks are proven to outperform conventional RKHS in many areas, because it is a non-linear function class with learnable features, that can work with uneven subspaces. The RBF kernel uses the same smoothing operator for all gradients. In Figure 3, we have shown that the RKHS is incapable of capturing functional gradient near connected clusters, while neural networks can do so. As a result, the sampling quality can be improved.

---

> > ### Comment · Reviewer_9msK · 2022-11-23
> > **Reply to authors**
> >
> > Thanks for answering my questions. You address all my questions. I will improve the score to 8.

---

> > > ### Author Response · Authors · 2022-11-24
> > > **Reply**
> > >
> > > We would like to thank you again for your effort and positive feedback! Your endorsement is very important to us. We are very happy that our response and updated manuscript have resolved your questions. Your valuable comments help us a lot and make our paper more readable and clear.
> > >
> > > Hope we can have more dicussions in the future and we believe that the discussions are helpful for the development of the community.
> > >
> > > Cheers,
> > >
> > > Authors

---

> ### Author Response · Authors · 2022-11-09
> **Response to Reviewer m3yE (Part 2/2)**
>
>
> **Q2: Preconditioned variants of Langevin dynamics.**
>
> We have several comparisons with the plain SGLD [1] and PFG also outperforms preconditioned SGLD in Table 2 [3]. Note that [3] is a representative version of [2], which is widely used in Bayesian neural network (BNN) literature with sufficient public implementations. Thus, we choose [1,3] as our competitor.
>
> [4] is a convergence analysis for Proximal Langevin Algorithm, which focus on the theory  rather than algorithmic issues. Besides, the proposed algorithm needs full gradient. Thus, it is not a direct competitor to our paper, but it is still insightful and we are glad to discuss it.
>
> Finally, we'd like to emphasize that our proposed algorithm is a variant of particle-based variational inference, which is deterministic and repulsive among particles. Although most  particle-based variational inference papers are restricted to RKHS space, we generalize the form and propose a regularization-based framework. The preconditioned one is a special and effective choice. Thus, rigorously speaking, SGLD variants are not direct competitors, but we think it is worthwhile to make such a comparison to justify the significance of pariticle-based VI framework so that we include them in our empirical study.
>
>
> [1] Max Welling and Yee Whye Teh. Bayesian Learning via Stochastic Gradient Langevin Dynamics.
>
> [2] Gaétan Marceau-Caron and Yann Ollivier. Natural Langevin Dynamics for Neural Networks.
>
> [3] C. Li et al., Preconditioned Stochastic Gradient Langevin Dynamics for Deep Neural Networks.
>
> [4] A. Wibisono. Proximal Langevin Algorithm: Rapid Convergence Under Isoperimetry
>
>
> **Q3: Why $-c\nabla^2\log p_*$ is better preconditioning?**
>
> As we mentioned, the Hessian inverse is a better preconditioning in optimization literature, which allows larger learning rates in a discretized algorithm. This is also the motivation of Newton's method and other second-order optimization algorithms.
>
> In sampling literature, the evolution of the posterior mean is equivalent to the conventional optimization. Thus, by extending the conclusion from optimization, the convergence of mean with $-c\nabla^2\log p_*$ preconditioner is better. In our Example 1 and Appendix A.6, it is also shown that the preconditioner improves significantly in Gaussian case.
>
> Besides, many preconditioned versions SVGD also adapts the Hessian inverse as the preconditioner (Appendix B.2). Unfortunately, the limitations of SVGD have already been justified by our analysis and their efforts are not perfect due to the RKHS restriction.
>
>
> **Q4: Is there a connection between Assumption A3 and weak learning conditions for boosting methods?**
>
> In conventional boosting algorithms, the function class of weak learners is usually small, so the capacity is insufficient, but the aggregated predictor performs well.
>
> In our algorithms, we also make restrictions on the gradient function class, which lead to a tractable sampling algorithm and the sampling performance can also be guaranteed.
>
> In a word, both conventional boosting and our algorithm are based on restricted function classes to obtain a good performance.

---

> ### Comment · Reviewer_m3yE · 2022-12-01
> **Thanks.**
>
> Thanks for your detailed explanations. I was convinced of the importance of the method. And I appreciate the author's effort to revise the manuscript. I would like to raise the score.

---

> > ### Author Response · Authors · 2022-12-02
> > **Thanks.**
> >
> > Thanks again for your helpful comments and positive feedback. Your suggestions are very important to us. We really appreciate that our response and updated manuscript have resolved your questions. Your valuable comments help us a lot to clarify some important parts.
> >
> > Hope we can have more discussions in the future and we believe that the discussions are beneficial to the development of the community.
> >
> > Cheers,
> >
> > Authors

---

### Official Review · Reviewer_9msK · 2022-11-03

**Confidence:** 5
**Clarity, Quality, Novelty And Reproducibility:** The quality of this paper is good.
**Correctness:** 4
**Technical Novelty And Significance:** 3
**Empirical Novelty And Significance:** 3
**Recommendation:** 8

**Strength And Weaknesses:**

Strength: The authors propose general metric space-induced gradient descent to design sampling algorithms. The generality of metrics (Q) may improve the sampling algorithm.

Weakness:

1. No convincing examples are provided analytically. I suggest that the authors study a Gaussian target distribution. In this case, the author may illustrate the advantage of general metrics and gradients.

2. The authors miss many crucial works in this field.  Please illustrate them in the context.

A. Garbuno-Inigo, F. Hoffmann, W.C. Li, A.M. Stuart. Interacting Langevin Diffusions: Gradient Structure And Ensemble Kalman Sampler. SIAM Journal on applied dynamical system, 2019.

W.C. Li, L.X. Ying. Hessian transport gradient flows. Research in the Mathematical Sciences, 2019.

Y.F. Wang, W.C. Li. Information Newton flow: second-order optimization method in probability space. 2020.

Y.F. Wang, W.C. Li. Accelerated Information Gradient flow.

Y.F. Wang, P. Chen, W.C. Li. Projected Wasserstein gradient descent for high-dimensional Bayesian inference.

Y.F. Wang, P. Chen, M. Pilanci, W.C. Li. Optimal Neural Network Approximation of Wasserstein Gradient Direction via Convex Optimization.

W.C. Li, A.T. Lin, G. Montufar. Affine Natural Proximal Learning. GSI, 2019.

A.T. Lin, W.C. Li, S. Osher, G. Montufar. Wasserstein Proximal of GANs. GSI, 2021.


**Summary Of The Paper:**

The paper studies the general gradient flow of KL divergences. The authors consider a generalized metric space and formulate the gradient descent as a least square problem. The metric spaces include L^2, RKHS, and covariance-weighted norms. Some linear and nonlinear families of functions, including neural networks, are used in approximating generalized metric spaces and gradients. Numerical examples are shown the effectiveness of the proposed methods.

**Summary Of The Review:**

The paper is written well with clear mathematics. Some analytical examples and important literature are missed.  It will take effect my ratings.

---

> ### Author Response · Authors · 2022-11-09
> **Response to Reviewer 9msK**
>
> Thanks for your suggestions and your insightful suggestions can indeed improve our work.
>
> **Q1. Analytical examples.**
>
> Thanks for your suggestion. In fact, we have included these materials in the Appendix, and we will highlight them in our current version. For Gaussian target distributions, our analytical results are provided in Appendix A2, which demonstrate that SVGD with standard kernels is not as good as our proposed algorithm. Moreover, Appendix A6 includes discretized convergence analysis to demonstrate the improvement over the plain version.
>
> We can summarize them briefly below:
>
> *1. SVGD with linear class*
>
> $
>  g(t,x)=-\Sigma_\ast^{-1}[(\Sigma_{t}+(\mu_{t}-\mu_{*})\mu_{t}^{\top})Kx + \mu_t - \mu_\ast] +K x.
> $
>
> Due to the $\mu_* \mu_t^\top$ term in the functional gradient, the distribution evolution will be unnecessarily rotated.
>
> *2. SVGD with RBF class*
>
> $
>  g(t,x) = O(x\exp(-\Vert x\Vert^2)).
> $
>
> It is a misspecified function class. The gradient in the tail part will diminish.
>
> *3. Linear function class with $L_2$ regularization $Q(x) =\frac{1}{2} \mathbb{E}\Vert x\Vert_2^2$*
>
> $
> g(t,x) = - \Sigma_\ast^{-1}(x-\mu_*)+\Sigma_t^{-1}(x-\mu_t).
> $
>
> It is a proper gradient without other information. However, when $-\nabla^2 \log p_\ast$ ($\Sigma^{-1}_\ast$ in Gaussian case) is highly ill-conditioned, the convergence can be slow.
>
> *4. Linear function class with Mahalanobis regularization $Q(x) = \frac{1}{2} \mathbb{E} \Vert x\Vert_{\Sigma_\ast^{-1}}^2$*
>
> $
> g(t,x) = -
>   (x-\mu_*)+\Sigma_*\Sigma_t^{-1}(x-\mu_t).
> $
>
> It includes the second-order information. When $-\nabla^2 \log p_*$ ($\Sigma^{-1}_*$ in Gaussian case) is highly ill-conditioned, the convergence is much faster than a non-preconditioned one.
>
> Our proposed trajectory is the shortest (Figure 2 (d)) that mimics the optimal transport path. Also, by Appendix A6, the discretized convergence analysis (for Gaussian case) also shows the superiority of our algorithm.
>
>
> **Q2. References.**
>
> Thanks a lot for your references. We have discussed and cited them in current version.

---

### Decision · Program_Chairs · 2023-01-20

**Decision:**

Accept: poster

**Justification For Why Not Higher Score:**

As stated in the summary tab, this paper has some weakness (1) lack of thorough theoretical justification, (2) weak comparison with existing methods such as Metropolis Adjusted Langevin Algorithm, and (3) weak numerical experiments. Although this study is novel and significant, this paper would not be appropriate for spotlight due to these weakness.

**Justification For Why Not Lower Score:**

The proposed algorithm is sufficiently novel. Introducing the preconditioning actually improves the performance. The theoretical justification is solid, and the paper is well written. Hence, this paper deserves acceptance.

**Metareview: Summary, Strengths And Weaknesses:**

This paper proposes a new particle based variational inference method that utilizes a preconditioning. The method estimates the gradient direction in the continuity equation by minimizing an L2 distance between the best direction with a preconditioned quadratic loss. The authors show the convergence of the algorithm in the infinite particle limit and the continuous time setting. The proposed method is examined by numerical experiments and its effectiveness is justified.

The proposed method is indeed new, and the theoretical justification is solid. The particle based variational method is one of the important issues in the literature. This paper gives a new insight to the literature. Overall, the paper is well written.
On the other hand, there are also some limitations. Convergence of discrete time dynamics and finite particle discretization is not given. The choice of H is not thoroughly exploited. Moreover, the motivation to utilize the neural network is not justified in a fully convincing way. Although the authors gave some additional remarks on the comparison with Langevin dynamics approaches, its comparison is not thorough. The author also stated that the Metropolis-Hastings correction is ad-hoc while it has a proper theoretical justification. However, this method lacks the convergence for discrete time and finite particle approximations. In that sense, comparison to Langevin type MCMC methods is not fully convincing. Finally, the experiments are conducted only on a simple setting.

However, this paper gives a nice contribution to the literature of the deterministic particle based variational method. The proposed method is sufficiently novel. Hence, I think it can appear in ICLR.





**Note From Pc:**

if the above contains the word "oral" or "spotlight" please see: "oral" presentation means -> notable-top-5% and "spotlight" means -> notable-top-25%. As stated in our emails, we are disassociating presentation type from AC recommendations